# Development of HYPER-P: HYdroclimatic PERformance-enhanced Precipitation at 1 km/daily over the Europe-Mediterranean region from 2007 to 2022

Paolo Filippucci[1], Luca Brocca[1], Luca Ciabatta[1], Hamidreza Mosaffa[1 2], Francesco Avanzi[3], Christian Massari[1]

[1]National Research Council, Research Institute for Geo-Hydrological Protection, Perugia, 06128, Italy
[2] Department of Geography and Environmental Science, University of Reading, United Kingdom
[3]Cima Research Foundation, Savona,17100, Italy

*Correspondence to*: Paolo Filippucci (paolo.filippucci@cnr.it)

**Abstract.** Accurate precipitation estimates are essential for a wide range of applications, including climate research, water resource management, agriculture, and natural hazard assessment. However, developing high-quality, long-term daily datasets at fine spatial resolutions remains challenging due to the inherent variability and heterogeneity of precipitation patterns. This study introduces the HYdroclimatic PERformance-enhanced Precipitation (HYPER-P) product, covering Europe and part of the Mediterranean basin from 2007 to 2022 at a 1 km daily resolution. HYPER-P is derived by downscaling and merging multiple data sources, including remote sensing products from Top-Down (TD) and Bottom-Up (BU) approaches, reanalysis datasets, and gridded in situ observations. The downscaling leverages on CHELSA climatology data, while the merging is obtained through a weighted average approach informed by Triple Collocation Analysis.

Four merged products were developed based on multiple combinations of satellite products, observation and reanalysis datasets. The evaluation of these products was conducted through high-resolution validation in three Mediterranean regions with dense observational networks and coarse-resolution validation across Europe and a portion of North Africa. Results indicate that the combination of TD and BU satellite approaches enhance precipitation estimates, with merged products outperforming the parent datasets, especially in regions with sparse gauge coverage. The inclusion of ERA5-Land further improves accuracy over areas characterized by complex topography. The merging of satellite products, particularly the one including ERA5-Land, shows overall strong performance, although challenges remain in validating precipitation estimates where ground observations are limited. This work contributes to advancing precipitation monitoring capabilities, offering valuable tools for scientific and operational applications across Europe and beyond.

**Keywords:** Precipitation, high-resolution, merging, triple collocation, satellite, Earth Observation

**Acronyms:**

ASCAT – Advanced Scatterometer

ASTER – Advanced Spaceborne Thermal Emission and Reflection Radiometer

GDEM – Global Digital Elevation Model

BU – Bottom-Up

CHELSA – Climatologies at high-resolution for the Earth's land surface areas

CHIRP – Climate Hazards Group InfraRed Precipitation

CHRS – Center for Hydrometeorology and Remote Sensing

CPC – Climate Prediction Center

DPR – Dual-frequency Precipitation Radar

EMO – European Meteorological Observations

EMS – Emergency Management Service

ERA5 – European Centre for Medium-Range Weather Forecasts, ECMWF, Reanalysis 5th Generation

EUMNET – European National Meteorological Services

E-OBS – ENSEMBLES daily gridded observational

GEO – Geostationary Orbit

GPM – Global Precipitation Measurement Mission

GSMAP – Global Satellite Mapping of Precipitation

H SAF – Satellite Application Facility on support to Operational Hydrology and Water Management

IMERG-ER – Integrated Multi-satellitE Retrievals for GPM - Early Run

IMERG-LR – Integrated Multi-satellitE Retrievals for GPM - Late Run

IMERG-FR – Integrated Multi-satellitE Retrievals for GPM - Final Run

IR – Infrared

JAXA – Japan Aerospace Exploration Agency

LEO – Low Earth orbit

M1 – Merge 1

M2 – Merge 2

M3 – Merge 3

M4 – Merge 4

MCM – Modified Conditional Merging

MW – Microwave

NHMS – National Meteorological and Hydrological Services

NOAA – National Oceanic and Atmospheric Administration

OPERA – Operational Program for Exchange of Weather Radar Information

PERSIANN – Precipitation Estimation from Remotely Sensed Information using Artificial Neural Networks – Cloud Classification System

R – Linear Pearson Correlation

RMSE – Root Mean Square Error

SAIH – Spanish Automatic System of Hydrologic Information

SM – Soil Moisture

SM2RAIN – Soil Moisture to Rainfall

SNR – Signal to Noise Ratio

SPEs – Satellite Precipitation Estimates

SRM – Snow Multidata Mapping and Modeling

TC – Triple Collocation

TD – Top-Down

VIS – Visible

USGS – United States Geological Survey

WSL – Swiss Federal Institute for Forest, Snow and Landscape Research

## 1 Introduction

Precipitation estimates are crucial in many fields of research such as climate studies (Pendergrass et al., 2017), water cycle research (Pellet et al., 2024), droughts (Serrano et al., 2010), floods (Maggioni and Massari, 2018), landslides (Peiro et al., 2024; Smith et al., 2023), ecosystem dynamic (Huxman et al., 2004), agriculture (Beck et al., 2020; Ru et al., 2022) and water resource management (Camici et al., 2024; Fischer and Knutti, 2016; Kucera et al., 2013). However, finding a high-quality, long-term daily precipitation at kilometer spatial resolution dataset is not straightforward, especially over data scarce region and complex terrain, given the spatial heterogeneity and temporal variability in precipitation.

In Europe and in the Mediterranean region, many observational, satellite-based and reanalysis datasets are available. Notable examples of ground-based datasets are E-OBS (Haylock et al., 2008; Cornes et al., 2018) and EMO-5 (Gomes et al., 2020) for the whole of Europe, and other regional datasets like SAFRAN for France, Spain, and Tunisia (Quintana-Seguí et al., 2008; Vidal et al., 2010; Quintana-Seguí et al., 2017; Tramblay et al., 2019) or MCM for Italy (Sinclair and Pegram, 2005; Bruno et al., 2021). These datasets, that rely mainly on rain-gauge observations, are the most reliable and widely used tool for directly measuring precipitation in Europe. However, they are often characterized by uneven distribution of ground monitoring networks, like, for example, the case of the Mediterranean region compared to Europe (Girons lopez et al., 2015), thus leading to potentially significant interpolation errors. Radar measurements are also increasingly available in Europe, with more than 200 operational weather radars managed by the EUMETNET within the OPERA (Huuskonen et al., 2014). This technology allows for the collection of reliable precipitation information with high temporal resolution (often in the range of minutes), wide spatial coverage (a single weather radar can cover a circular area with a radius of 100-250 km) and better spatial accuracy

compared to rain gauges, which only measure precipitation at their exact location). Radars do not measure rainfall directly; instead, they detect the reflectivity of precipitation particles. Radar measurements are hence often combined with rain gauges, to adjust the measurements and obtain more reliable precipitation estimates. However, also this network is heterogeneous in hardware, signal processing, frequency and scanning strategy, therefore their combination is difficult and prone to errors. Moreover, most of the existing weather radars are mainly located in developed countries (Heistermann et al., 2013).

Satellite and reanalysis datasets are widely used alternatives to overcome the problem of spatial accuracy variations. The Global Precipitation Measurement (GPM) mission, launched in 2014 by NASA and JAXA in collaboration with GES DISC, revolutionized precipitation retrieval with a multi-sensor integration approach (Hou et al., 2014). GPM intercalibrates, merges, and interpolates data from various instruments to generate half-hourly precipitation estimates on a 0.1° grid across the 60° N–S domain via the Integrated Multi-Satellite Retrievals for GPM (IMERG; Huffman et al., 2018). It offers three Level 3 products with different timeliness and calibration approaches: IMERG-ER, IMERG-LR, and IMERG-FR. While IMERG employs a Top-Down (TD) approach, based on the inversion of the atmospheric signals to obtain instantaneous rainfall rates, other satellite rainfall products developed in the last decade are based on indirect and alternative approaches. For example, SM2RAIN-ASCAT (Brocca et al., 2014; 2019) is obtained with a Bottom-Up (BU) approach, i.e. by inverting the soil water balance equation for rainfall (only the liquid phase of precipitation) starting from satellite soil moisture observations derived from ASCAT SM data (H SAF, 2020). Rainfall data are generated for the entire terrestrial globe, excluding frozen areas, and are available on a 0.1° grid.

While invaluable, SPEs face various challenges. Specifically, precipitation products obtained from the Top-Down approach have limitations related to the instantaneous nature of the measurement (related to the satellite overpass) with respect to the sporadic nature of natural phenomenon, leading to errors influenced by precipitation type, satellite orbit, and swath width (Kucera et al., 2013; Behrangi and Wen, 2017). Additional issues include biases, difficulties in light precipitation estimation, and detection over snow and ice (Ferraro et al., 1994; Kidd and Levizzani, 2011). The GPM's DPR has mitigated some of these issues, but further improvements are needed (Tan et al., 2016; Gebregiorgis et al., 2018). Similarly, alternative approaches, like SM2RAIN, have also some limitations, due to e.g. the underestimation of the rainfall when the soil saturates, the low accuracy of SM (and therefore rainfall) for dense vegetation coverage and complex topography) and the sensitiveness to SM variations induced by noise (Brocca et al., 2014, 2019).

Model reanalysis datasets like ERA-Interim and ERA5 (Dee et al., 2011; Hersbach et al., 2020, Munoz Sabater et al., 2019) provide an alternative to satellite and ground-based precipitation data. While effective for simulating large-scale weather patterns, their low spatial resolution and limitations in sub-grid process parameterization hinder accurate representation of convective systems (Ebert et al., 2007). In other words, reanalysis and satellite datasets are widely used alternative but also suffer from uncertainty especially over mountainous regions (Maggioni et al, 2018, Gomis-Cebolla et al., 2023) and challenge to detect small scale precipitation patterns typical of complex landscapes such as the Alpine region (Girotto et al., 2024).

Recently, some studies have proposed merged precipitation datasets as an alternative to single source estimates (Beck et al. 2019, Pellarin et al. 2013, Massari et al. 2020) which, thanks to optimal merging, have shown to overcome the problems of parent datasets (Beck et al 2017, Brocca et al. 2020, Camici et al. 2018). Still the spatial resolution of these products remains quite coarse relative to the fine landscape features of the European regions.

The aim of this paper is to present the HYdroclimatic PERformance-enhanced Precipitation (HYPER-P) product available over Europe and part of the mediterranean basin from 2007 to 2022 at 1 km/daily spatial and temporal resolution, as well as its quality and its potential usage.

This product is generated by downscaling and merging multiple precipitation datasets from different sources: rain gauges, satellite observations (using both top-down and bottom-up approaches), and reanalysis data. The parent datasets are selected based on criteria such as low latency availability or potential, broad spatial coverage, and high accuracy. As a result, the merged product can be made available globally with relatively short latency—approximately one week. Radar measurements are not included in the merging process due to the lack of a global radar dataset and the limited number of weather radars, particularly in developing countries, but they can be used as a valuable reference. Local (intended as not-global) datasets from radar and gauge were not included in the merging, but they were used as independent references for assessing the performance of the merged product. The parent products are first downscaled using monthly pattern information obtained by CHELSA climatology dataset and then merged using a weighted average where relative weights have been calculated based on the relative quality derived from TC Analysis (Gruber et al. 2016; 2017, Massari et al. 2017, Chen et al. 2021). Due to its potential low latency, HYPER-P can be useful for climatological applications like hydrological modeling, agricultural and drought monitoring or climatological studies. Specifically, HYPER-P is expected to be particularly valuable for completely or nearly ungauged areas, which lack stable and high-resolution information from ground networks (gauges and/or radars). The evaluation of the dataset is carried out at both high- and coarse-resolution. The high resolution is based on high-density ground-based rainfall networks over three different regions, while the coarse one is obtained through the comparison against gridded precipitation datasets available over Europe as E-OBS and EMO. Various satellite precipitation products are also evaluated in the same areas for comparison.

This paper is organized as follows: after the introduction, the study area and the datasets used in this study are described. Subsequently, the downscaling and merging procedures are explained in detail, along with the performance metrics adopted for product assessment. Next, high-resolution validation is performed at 1-km spatial resolution over three sub-regions of the Mediterranean Basin with dense rain gauges and/or radar networks. This is followed by a broader coarse-resolution analysis across the full domain, using the highest-performing products aggregated to 10 km resolution. Finally, the results of the two validations are utilized to assess the merged products and evaluate their validity across the European region and a portion of north Africa regions.

## 2 Data

### 2.1 Study Area

The target area of this study is the Europe and mediterranean regions, specifically the area between -11.9° and 44.8° East and 27.7° and 62.3° North. The Europe region represent diverse climatic zones, which play a crucial role in precipitation dynamics. Specifically, the following climates are found in different areas: Oceanic Climate (Cfb), found in Western Europe, particularly along the Atlantic coast, this climate is characterized by mild temperatures and relatively high, evenly distributed precipitation throughout the year; Continental Climate (Dfb, Dfc), dominates Eastern and Central Europe, marked by greater seasonal temperature variation, with cold winters and warm summers, and moderate precipitation, often peaking in summer; Mediterranean Climate (Csa, Csb), prevails in Southern Europe, particularly around the Mediterranean Basin. It features hot, dry summers and mild, wet winters; Subarctic and Polar Climates (Dfc, ET), present in Northern Europe and high-altitude regions, such as Scandinavia and parts of the Alps, with cold winters, short summers, and generally low precipitation, even if the orographic lift in some cases affect the precipitation pattern (Bonacina et al., 1945). These regions provide distinct hydrological contexts, with variability in precipitation driven by geographical, seasonal, and synoptic-scale atmospheric processes.

### 2.2 Dataset selections

Several precipitation datasets have been downloaded and processed to create a more reliable high-resolution precipitation product over the study area and validate it. Specifically, we collected seven reference datasets: four high-resolution (MCM, SAFRAN, COMEPHORE and EMO) and three medium resolution reference datasets (CPC, E-OBS and ERA5-Land). These datasets were used to validate the coarse-resolution satellite from the bottom-up and top-down approaches (CHIRP, SM2RAIN-ASCAT, IMERG-LR, CPC), their downscaling and the merged products. The following section provides a detailed description of these, and all the other datasets used in the analysis. The details of each dataset are reported in Table 1.

**Table 1: characteristics of the precipitation dataset selected for intercomparison: name, temporal resolution, spatial sampling, spatial coverage, period availability, and source (satellite TD: Top-down approach, satellite BU: Bottom-up approach)**

| | Dataset | Temporal resolution | Spatial sampling | Spatial coverage | Period | Source |
|---|---|---|---|---|---|---|
| REFERENCE | COMEPHORE | Hourly | 0.009° | Hérault Basin | 1997-2021 | Reanalysis (Gauge+Radar) |
| | CPC | Daily | 0.5° | World (Land) | 1979-date | Gauge |
| | EMO | Daily | 1 arcmin | Europe | 1990-2022 | Gauge, reanalysis |
| | E-OBS | Daily | 0.1° | Europe | 1950-2021 | Gauge |
| | ERA5-Land | Hourly | 0.1° | World (Land) | 1950-date | Reanalysis |
| | MCM | Daily | 0.009° | Po basin | 2016-2021 | Gauge, radar |

| | | | | | | |
|---|---|---|---|---|---|---|
| | SAIH | Daily | 0.009° | Ebro Basin | 1987-date | Gauge |
| SATELLITE | CHIRP | Daily | 0.05° | 50 °S : 50°N | 1981-date | Satellite TD |
| | GSMAP | Daily | 0.1° | 60 °S : 60 °N | 2003-date | Satellite TD |
| | IMERG Late Run | Half-hourly | 0.1° | World | 2000-date | Satellite TD |
| | PERSIANN | Daily | 0.04° | 60 °S : 60 °N | 2003-date | Satellite TD |
| | SM2RAIN-ASCAT | Daily | 12.5 km | World (Land) | 2007-2023 | Satellite BU |

### 2.2.1 Reference datasets

**COMEPHORE**: Comephore (Tabary et al. 2012) precipitation reanalysis is available between 1997 and 2021. The precipitation estimates are obtained using the data from the French operational weather radars network ARAMIS, corrected by hourly rain gauges observations interpolated by kriging (around 4,000). The COMEPHORE product has a spatial resolution of 1 km and a temporal resolution of 1 hour over the Hérault basin (~18,000 km$^2$). The hourly precipitation was accumulated to daily scale in this study.

**CPC**: This dataset is part of products suite from the CPC Unified Precipitation Project that are underway at NOAA CPC. The dataset is obtained by combining all gauge information sources available at CPC (around 17,000) and by taking advantage of the optimal interpolation objective analysis technique (Xie et al., 2007). See Chen et al. (2008), for further details. Precipitation data are available with a spatial resolution of 0.5° latitude x 0.5° longitude. The daily precipitation product was provided by the NOAA PSL, Boulder, Colorado, USA, from their website at https://psl.noaa.gov. This product is available with low latency (around three days).

**EMO:** EMO is a European high-resolution, daily meteorological dataset built on historical and real time observations developed within Copernicus EMS. Among the released variables, the product provides total precipitation. The insitu observations are quality checked and then interpolated through SPHEREMAP and Yamamoto methods (Gomes et al., 2020). Currently, EMO is available in two spatial resolutions: EMO-5 provides grids with a spatial resolution of 5kmx5km and covers the period from 1990 to 2019. EMO-1 provides grids with a spatial resolution of 1arcminx1arcmin (approx. 1.5km) and covers the period from 1990-2022.

**E-OBS:** E-OBS is a land-only gridded daily observational dataset for precipitation and other meteorological variables in Europe. This dataset is based on observations from meteorological stations over Europe (14,212 stations with data after 2007 in version 28e), which are provided by the NMHSs and other data holding institutes (Cornes et al., 2018). The product is available at daily temporal resolution and 10 km spatial resolution. Version 25.0e was adopted for this study. Note that, in some areas, E-OBS observations are derived by aggregating precipitation networks with time intervals that differ from the standard 00–24 period (Overeem et al., 2023), This can potentially cause uncertainty on the assessment precipitation products

using E-OBS. However, considering that E-OBS is not the only dataset used as reference and the importance of assessing HYPER-P against widely used precipitation products, the uncertainty is deemed acceptable.

**ERA5-Land:** ERA5-Land, provides hourly data of various land surface variables from 1950 onwards, combining models with observations. It was produced by replaying the land component of the ECMWF ERA5 climate reanalysis and it is characterized by an improved spatial resolution (0.1 degree), while the temporal resolution is 1 hour (Hersbach et al., 2020, Munoz Sabater et al., 2019). The hourly precipitation was accumulated to daily scale in this study.

**MCM:** High-resolution precipitation fields over the Po basin (around 80,000 km$^2$) were estimated with the MCM technique, which incorporates precipitation gauges and radar estimates to infer 1 km precipitation observations at hourly time scale (Bruno et al., 2021). MCM is an improvement of the Conditional Merging proposed by Sinclair and Pegram (2005), which estimates the structure of covariance and the length of spatial correlation at every gauge, taking it from the cumulated radar precipitation fields. For the Po River basin, MCM is based on 1,377 precipitation gauges and on the mosaic of the Italian weather radars. This product has been developed and shared by the CIMA Research Foundation.

**SAIH:** A high-resolution forcing dataset of precipitation based on the SAFRAN analysis system (Quintana-Seguí 2016, 2017) has been created by the Ebro Observatory for the Ebro basin area (~83,000 km$^2$). This dataset uses in-situ data from the SAIH and includes precipitation data gathered from 333 stations over the Ebro basin every 15 minutes. There are several versions of this dataset at different resolutions. The 1 km resolution product has been created specifically for the ESA 4DMED-Hydrology project. It covers the period 2008-2020.

### 2.2.2 Satellite datasets

**CHIRP:** The CHIRP is a quasi-global precipitation data set. The product uses IR data to retrieve precipitation at high-resolution (Funk et al., 2015; Shen et al., 2020). The data set runs from 1981 to the near present. The CHIRP satellite was developed by the USGS in collaboration with the Climate Hazards Group at the University of California. CHIRP combines Thermal Infrared satellite precipitation estimates from the Globally Gridded Satellite (GriSat) and the Climate Prediction Center dataset (CPC TIR) from NOAA to produce the precipitation dataset. It is therefore based on a TD approach, basing precipitation information from cloud and atmosphere measurements. The CHIRP product provides satellite estimates at high spatiotemporal resolution covering regions between 50° S to 50° N of latitudes. The selected version does not use any rain-gauge data.

**IMERG-LR:** The IMERG algorithm uses data obtained from GPM mission to estimate precipitation over the majority of Earth's surface (Huffman et al., 2019). The precipitation is obtained by exploiting the TD approach, where the precipitation

particles are sensed from different satellite sensors in various regions of the electromagnetic spectrum: VIS, IR, and MW. The resulting product spatial resolution is 0.1 degree, while the temporal resolution is 30 minutes. Here, the Late-run version 6 of the dataset is adopted, characterized by 12–18 h latency. In this study, the precipitation data were accumulated to obtain daily measurements. The selected version does not use any rain-gauge data.

**GSMAP:** The GSMAP is developed by JAXA. The product takes advantage of the GPM mission constellation satellites to provide hourly rain rates (Kubota et al., 2020). The precipitation estimation is based on the merging of microwave and infrared retrievals through LEO and GEO platforms. It relies on the TD approach. The product covers 60°S to 60°N globally. The selected version does not use any rain-gauge data.

**PERSIANN:** PERSIANN-CCS (here in after PERSIANN) is developed by the CHRS at the University of California, Irvine. The product is based on the use of neural network function procedures to compute precipitation rate at each 0.04° x 0.04° pixel through infrared images provided by geostationary satellites, hence it is based on the TD approach (Ashouri et al., 2015). The product covers 60°S to 60°N latitude. The selected version does not use any rain-gauge data.

**SM2RAIN-ASCAT:** This dataset is a new global scale rainfall product obtained from ASCAT satellite soil moisture data through the SM2RAIN algorithm (Brocca et al., 2014; 2019). This algorithm is based on the BU approach, since it infers the precipitation from SM variations by resolving the soil water balance equation. The SM2RAIN-ASCAT rainfall dataset (in mm/day) is provided over a regular grid at 0.1-degree sampling on a global scale. The product represents the accumulated rainfall between 00:00 and 23:59 UTC of the indicated day. The SM2RAIN method was applied to the ASCAT soil moisture product (Wagner et al., 2013), H SAF H119-H120 product, for the period from January 2007 to December 2021 (15 years). It is potentially available with low latency. The selected version does not use any rain-gauge data.

All the above precipitation data (reference and merged) were linearly interpolated each day at midnight UTC, with a maximum gap of empty data of 5 days.

### 2.2.3 Additional datasets

Two additional datasets were collected to obtain a water mask (DEM ASTER) and statistical information regarding high-resolution precipitation patterns (CHELSA dataset). Brief descriptions of them are provided below:

**CHELSA:** CHELSA (Karger et al., 2017) is a very high-resolution (30 arc sec, ~1km) global downscaled climate dataset currently hosted by the WSL. It is based on a mechanistical statistical downscaling of global reanalysis data or global circulation model output, and it includes climate layers for various time periods and variables. The precipitation algorithm incorporates orographic predictors including wind fields, valley exposition, and boundary layer height, with a subsequent bias correction. Monthly precipitation data from 2000 to 2019 were downloaded.

**ASTER GDEM**: Elevation data were obtained by the ASTER, one of five instruments aboard NASA's Terra spacecraft (launched in 1999). The ASTER GDEM covers land surfaces between 83°N and 83° with a spatial resolution of 1 arcsec (~30 m resolution at the Equator). Here, version 3 data was used to obtain the average DEM at 1 km scale and generate a water mask (NASA et al., 2018).

## 3 Methods

### 3.1 Downscaling Procedure

One product for each category (Gauge/Radar, Reanalysis, Satellite TD and BU) was selected for downscaling at 1 km spatial resolution, using CHELSA climate information. The products were selected with the criteria of maximum spatial coverage, lowest latency, higher spatial resolution and accuracy. The selected products were CPC for Gauge data, ERA5-Land for Reanalysis, IMERG-LR for satellite TD approach and SM2RAIN-ASCAT for satellite BU approach. A regular grid of 0.009° spatial resolution (1 km at equator) was created for the whole study area. Only land pixels are used for this analysis. A mask is derived from ASTER dem to mask water pixels.

A downscaling procedure is carried out for each precipitation product by using the high-resolution information contained in CHELSA. The downscaling procedure is developed starting from the work of Terzago et al. (2018). The main steps of the procedure are shown in Fig. 1. CHELSA data are extracted and bilinear interpolated on the chosen grid (because the CHELSA grid is different from the target one). Since CHELSA dataset is not available in real-time but only up to June 2019, a single standard year climatology was used for the full period: first, monthly aggregates were converted into average daily precipitation by dividing each of them by the number of days in the corresponding month. Then, CHELSA estimates for the same month across 2000-2019 were averaged to obtain 12 maps, each representative of a different month, thereby producing the standard year estimates (Fig. 1a). The average monthly values of the standard year were then attributed to the central day of each corresponding month of the study period. Linear interpolation was then applied to obtain a daily estimate across the entire study period, thus avoiding step patterns after the downscaling. CHELSA information is used just for the relative precipitation patterns (not the value itself), to spatialize the coarse-resolution information of the selected precipitation products. Indeed, the pattern information of CHELSA is derived from the modelling of orographic predictors of wind fields, valley exposition and boundary layer height and therefore can be considered a reliable estimation of the real precipitation distribution (Karger et al., 2017). The relative pattern is hence obtained by dividing the CHELSA precipitation amount by the results of a moving gaussian spatial filter applied to the same data. The moving gaussian filter is used to reproduce the parent product original spatial sampling, therefore its standard deviation is fixed to half the spatial sampling of the downscaled precipitation product, i.e. 5, 5, 6 and 25 km for ERA5-Land, IMERG, SM2RAIN_ASCAT, and CPC, respectively (Fig. 1b).

The precipitation datasets to be downscaled (Fig. 1c) are first resampled to the project grid through a bilinear interpolation, to exploit the spatial information of the original product at its fullest. A moving gaussian spatial filter with the same standard deviation of the previous is then applied to the interpolated data, to smooth the precipitation pattern obtained by the bilinear interpolation (Fig. 1d). The obtained filtered data are hence multiplied for the CHELSA-derived weights, obtaining a "pre-downscaled" product (Fig. 1e). Although this strategy does not allow a precise downscaling of the single storm pattern, due to the absence of a concurrent high-resolution pattern, it is useful to better spatialize coarse resolution information through the year.

The downscaling process applied at daily temporal resolution may introduce errors derived by the use of monthly CHELSA data. To mitigate these errors, a correction factor was applied to preserve the pattern of the parent data. This factor was obtained by first aggregating the pre-downscaled precipitation data at coarse resolution and then calculating the ratio between this aggregation and the original pre-downscaled precipitation. A moving gaussian spatial filter was applied to smooth transitions and avoid step behaviors (Fig. 1g). The downscaled data are finally obtained by multiplying the pre-downscaled data by the correction factor (Fig. 1h). The coarse-resolution aggregation of the downscaled data shows that the original coarse precipitation values are mainly maintained (Fig. 1i), since slight changes in precipitation are almost always present when spatial interpolation algorithms are applied.

The procedure ensures the reproduction of climatologically consistent monthly precipitation patterns using the CHELSA product, while preserving sub-monthly precipitation variability and maintaining the total precipitation at the coarse scale as evaluated by the parent product.

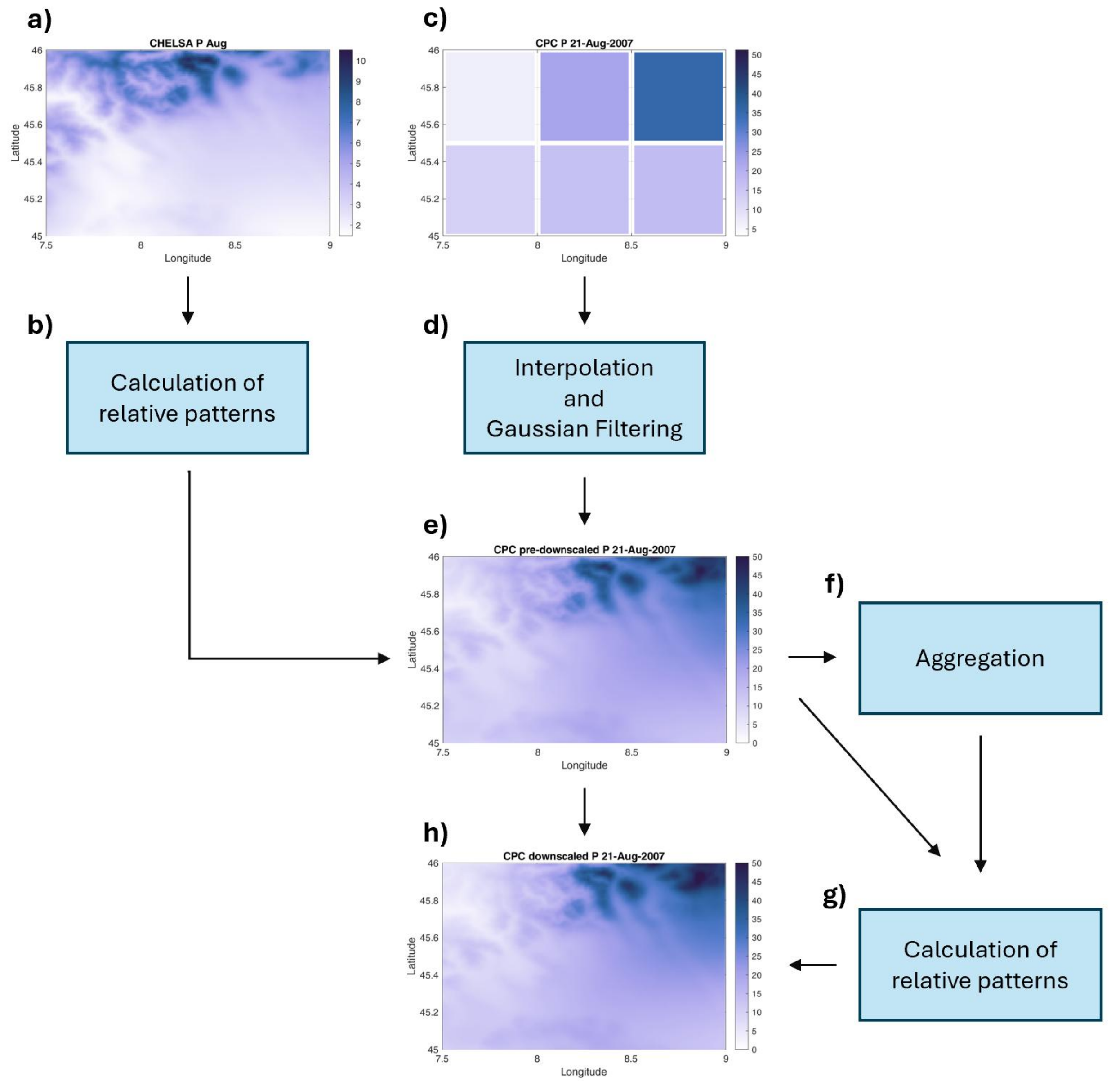

**Figure 1: Example of the downscaling procedure for CPC data of the 21 August 2007 in a small catchment of Italian Alps. a) original CHELSA precipitation pattern for August, b) relative CHELSA spatial pattern for the 21st of August, c) original CPC data, d) CPC data after spatial interpolation and gaussian filtering, e) pre-downscaled data, f) aggregation of the pre-downscaled data at the original coarse-resolution, g) corrective pattern of the pre-downscaled data, h) downscaled data, i) aggregation of the downscaled data at the original coarse-resolution**

### 3.2 Merging Procedure

The objective product should be available with a relatively low latency (e.g., a week), therefore ERA5-Land, CPC, IMERG-LR and SM2RAIN-ASCAT products are selected to be downscaled and merged, due to their low latency availability or potentiality, large coverage and accuracy. The following combinations of the datasets are here tested:

1) M1 = Gauge+Satellite TD;
2) M2 = Gauge+Satellite TD+Satellite BU;
3) M3 = Satellite TD+Satellite BU;
4) M4 = Reanalysis+Satellite TD+Satellite BU.

The comparison between M1 and M2 combinations allow us to assess the improvement related to the addition of the relatively new BU new approach to the merging procedure. Moreover, ASCAT SM has been available only since 2007, while CPC and IMERG both have data from 2000 onward. In M3 combination, a precipitation product derived only from satellite data is developed to assess the satellite capability to estimate precipitation also in absence of ground observations. Finally, the reanalysis product is used in place of the gauge one in M4 combination, to assess its potential, also considering the difference in spatial resolution of the selected products - ~50 and 10 km for the CPC and ERA5-Land products, respectively.TC technique (Gruber et al., 2016; 2017; Massari et al., 2017; Chen et al., 2021) is adopted to merge the different data. The approach requires three independent datasets with uncorrelated errors. The mechanism of TC approach is established on a linear error model, which can be represented by the equation as:

$$X_i = \alpha_i + \beta_i t + \varepsilon_i \tag{1}$$

where $X_i$ (i =1, 2, 3) are collocated measurement systems linearly related to the true underlying value $t$ with additive random errors $\varepsilon_i$, respectively, while $\alpha_i$ and $\beta_i$ are the ordinary least squares intercepts and slopes. By assuming that the errors from the independent sources have zero mean (E($\varepsilon_i$)=0) and are uncorrelated with each other (Cov($\varepsilon_i$, $\varepsilon_j$) =0, with i ≠ j) and with $t$ (Cov($\varepsilon_i$, $t$)=0) the variance of the error of each dataset can be expressed as (McColl et al., 2014):

$$\sigma_\varepsilon = \begin{cases} \sqrt{Q_{11} - \frac{Q_{12}Q_{13}}{Q_{23}}} \\ \sqrt{Q_{22} - \frac{Q_{12}Q_{23}}{Q_{13}}} \\ \sqrt{Q_{11} - \frac{Q_{12}Q_{23}}{Q_{12}}} \end{cases} \tag{2}$$

From the error variance, Gruber et al. (2016) obtained the SNR value of each dataset of the triplet with respect to the unknown truth for each pixel of the study area. This index can be considered a relative indicator of the capacity of the dataset to contain precipitation information with respect to the other two. An optimal merging of the products can be therefore obtained by

$$P_{Merged} = \omega_1 P_1 + \omega_2 P_2 + \omega_3 P_3 \tag{3}$$

with

$$\omega_1 = \frac{SNR_1}{SNR_1+SNR_2+SNR_3} \qquad \omega_2 = \frac{SNR_2}{SNR_1+SNR_2+SNR_3} \qquad \omega_3 = \frac{SNR_3}{SNR_1+SNR_2+SNR_3} \tag{4}$$

This approach was then applied to all the selected combinations. When just two datasets were selected for merging, ERA5-Land product was selected to complete the triplet (not used in the merging). In these cases, the merged precipitation can be obtained from equations 3 and 4 by ignoring the third product index (related to ERA5-Land):

$$P_{Merged} = \omega_1 P_1 + \omega_2 P_2 \tag{5}$$

with

$$\omega_1 = \frac{SNR_1}{SNR_1+SNR_2} \qquad \omega_2 = \frac{SNR_2}{SNR_1+SNR_2} \tag{6}$$

The obtained weights for all the four combinations are shown in Appendix A. SM estimates from space are unreliable when the soil is in frozen conditions. This reduces the applicability of the BU approach in frozen areas. Since SM2RAIN ASCAT data are derived from SM, precipitation estimates obtained in frozen conditions are excluded from the analysis. The frozen condition mask is obtained from ERA5-Land, by selecting all the dates in which the surface temperature of the first soil layer is below 0. In the masked areas, SM2RAIN ASCAT product cannot be used, hence the TC fails in obtaining the SNR. In these areas, three different approaches were tested, according to the analysed products: a) for M2 combination, in frozen conditions SNR was obtained by replacing SM2RAIN-ASCAT with ERA5-Land in the TC triplet, i.e. M1 data were used in frozen areas; b) for M3, the full weight is given to IMERG data when SM2RAIN-ASCAT is not available; c) for M4, the SNR of IMERG and ERA5-Land were interpolated from the nearest 4 valid pixels, and the precipitation was obtained by merging the two of them.

### 3.3 Performance metrics

The precipitation estimates assessment against the benchmarks data was carried out by calculating different metrics, specifically as follows:

- R expresses the linear relationship between two sets of data. It ranges between −1 and +1, where −1 indicates perfect negative linear relationship, +1 means perfect positive linear relationship and 0 means no statistical dependency. Pearson's correlation is here obtained from

$$R = \frac{\sum((P_{est} - \overline{P_{est}}) * (P_{obs} - \overline{P_{obs}}))}{\sqrt{\sum(P_{est} - \overline{P_{est}})^2 \sum(P_{obs} - \overline{P_{obs}})^2}}$$

  where $P_{est}$ and $\overline{P_{est}}$ are the daily precipitation estimates and the average precipitation estimates, respectively, while $P_{obs}$ and $\overline{P_{obs}}$ are the daily and average observed precipitation. This index was calculated both in space and in time. For the

spatial Pearson correlation, the precipitation was first accumulated at monthly temporal resolution, in order to match CHELSA original resolution.

- RMSE is a widely used index to measure the error between an estimated and an observed dataset. Three different sources of error are considered together: decorrelation, bias and random error. As the name implied, RMSE is obtained by calculating the square root of the mean quadratic difference between two datasets:

$$RMSE = \sqrt{\overline{(P_{est} - P_{obs})^2}}$$

- BIAS index measures the systematic over- or underestimation of one dataset with respect to the benchmark data. Here, it is obtained from the difference between the estimated and the observed precipitation. According to the above definition:

$$BIAS = \sum (P_{est} - P_{obs})$$

Negative BIAS values indicate precipitation underestimation, while positive bias values mean the opposite.

## 4 Results

### 4.1 High-resolution validation

In this section, the performances of all datasets against high spatial resolution reference data are presented and discussed. All datasets were linearly interpolated to the same 1 km grid using bilinear interpolation to enable comparison. The validation of the precipitation products was conducted in three selected study areas, where high spatial and temporal resolution observed datasets were available. Specifically, the validation was carried out in 1) the Po River basin (Italy), using MCM data as benchmark; 2) the Hérault basin (France), using the COMEPHORE precipitation reanalysis as benchmark; and 3) the Ebro basin (Spain), using the SAIH meteorological dataset as benchmark.

The results of the analysis are shown in Fig. 2-4. Among the satellite precipitation products, PERSIANN performed the worst for all metrics and across all study areas, followed by GSMAP and CHIRP. IMERG-LR and SM2RAIN-ASCAT demonstrated the best overall performances, with the former exhibiting higher temporal Pearson's correlation, and the latter achieving greater monthly spatial Pearson's correlation and lower RMSE. This confirms the selection of SM2RAIN-ASCAT and IMERG-LR for being merged within the integrated products, as they show superior performance and wider spatial coverage (CHIRP and PERSIANN are unavailable for high and low latitudes).

The assessment of reference products is less straightforward and site dependent: overall, CPC has the lowest performance, likely due to its coarse spatial resolution; while the performance of the remnant products depends on the study area. In the Po River basin (Fig. 2), ERA5-Land displayed the best Temporal Pearson's correlation and good monthly spatial Pearson's correlation, despite a general overestimation of precipitation (positive BIAS). In contrast, EMO and E-OBS tended to underestimate precipitation. In the Hérault River basin (Fig. 3), instead, ERA5-Land's performances remained mainly stable, while E-OBS exhibited the highest spatial correlation. Finally, in the Ebro River basin (Fig. 4), both EMO and E-OBS performed well: specifically, E-OBS showed a double-edged pattern in the violin plots of temporal correlation and RMSE, suggesting non-uniform performance across the region. Indeed, since both the benchmark (SAIH) and reference (E-OBS) products are derived from gauge data, it is probable that these discrepancies are related to a partial overlap in the gauge sensors included in the two products. This is strongly supported by the results shown in Fig. 5, which compares the temporal Pearson's correlation of EMO/E-OBS against observations with the distribution of their gauge networks in the three study areas. It is worth noting that the local benchmark gauge distribution is not shown here; however, the gauge locations in the Ebro region can be inferred from the patterns in the relative Pearson's correlation map, reflecting the Thiessen polygons used to create precipitation products; while the MCM network is available from Fig. 8 of Filippucci et al. (2022). Additionally, both MCM and COMEPHORE are derived from the integration of pluviometers and meteorological radar measurements, meaning their spatial capability for estimating precipitation is greater than what might be inferred from gauge locations alone, thanks to the large coverage of radar measurements and their high spatial resolution (<1 km). Regarding EMO and E-OBS, the highest performances in each study area were observed close to the gauges, indicating that they are likely used for the generation of both EMO/E-OBS datasets and the high-resolution local precipitation products. Despite this fact introducing bias in our analysis, these findings highlight that the performance of gauge-based products is closely linked to the density and distribution of the gauge stations.

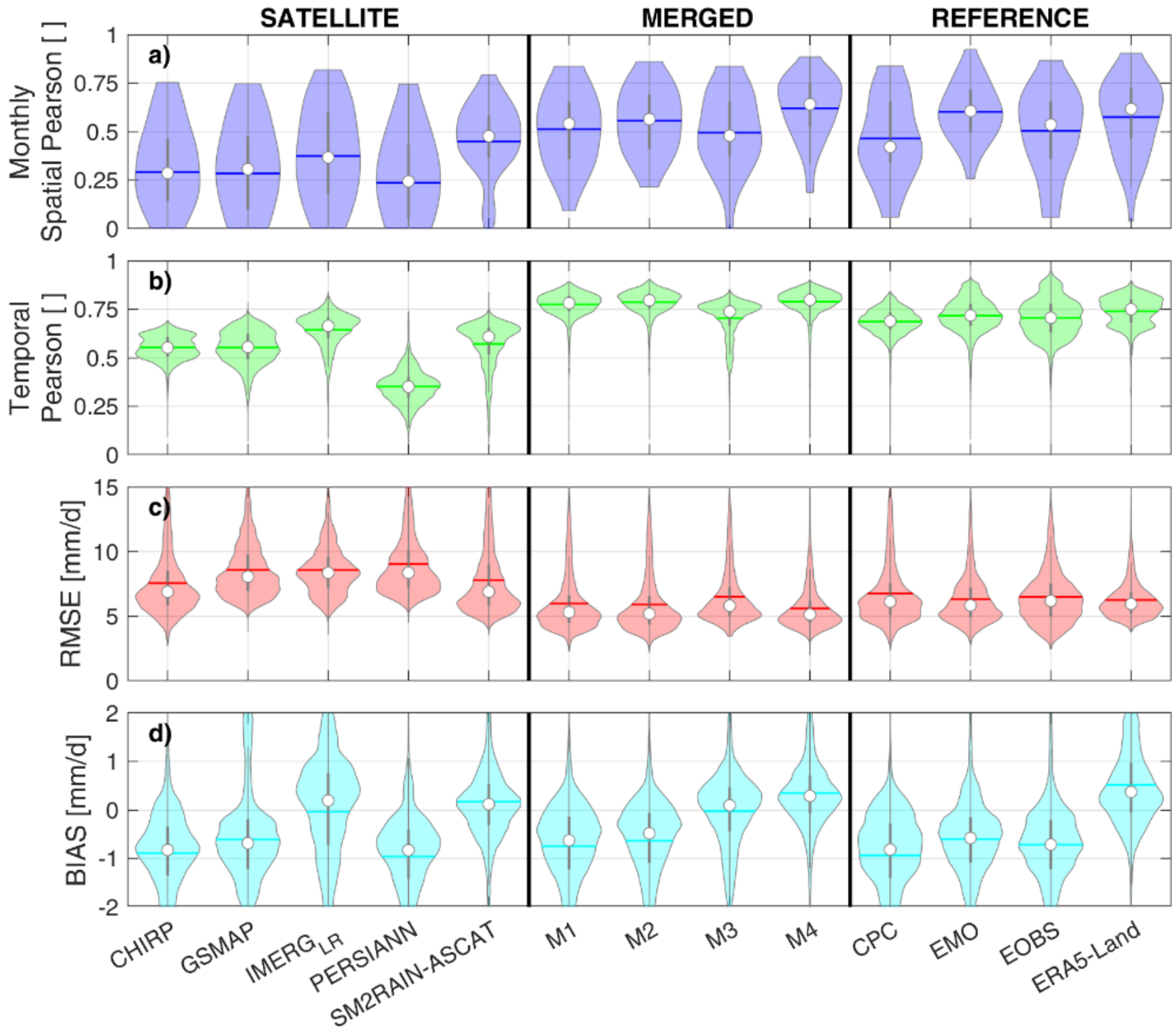

**Figure 2: Spatial Pearson's correlation at monthly scale, Temporal Pearson's correlation, BIAS error and RMSE at daily scale for the Po River basin against MCM benchmark (1 km spatial resolution). For each violin, the white dot is the average value, the dark line the median value, and the shape of the violin reflects the data distribution.**

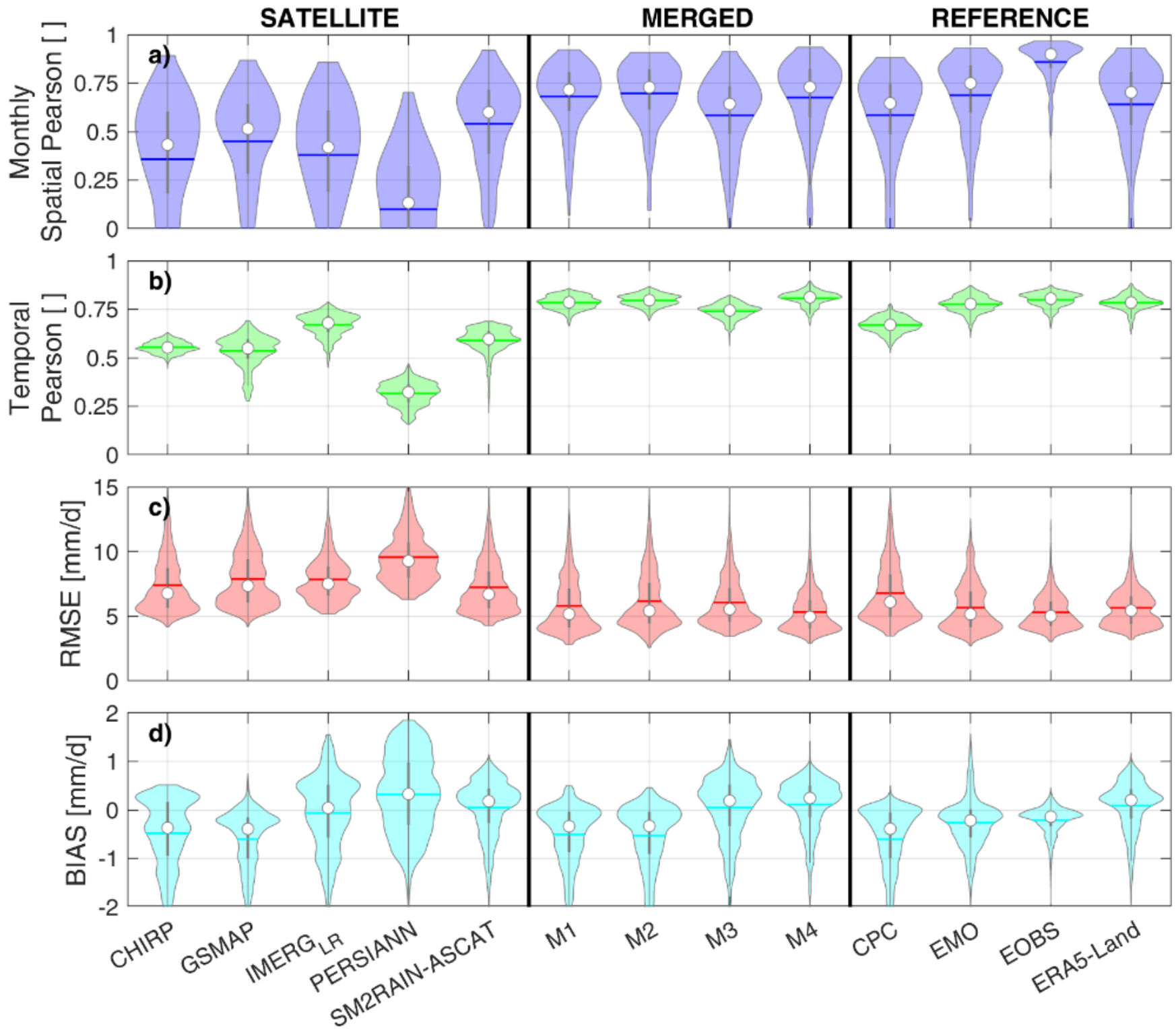

**Figure 3: Spatial Pearson's correlation at monthly scale, Temporal Pearson's correlation, BIAS error and RMSE at daily scale for the Hérault River basin against COMEPHORE benchmark (1 km spatial resolution). For each violin, the white dot is the average value, the dark line the median value, and the shape of the violin reflects the data distribution.**

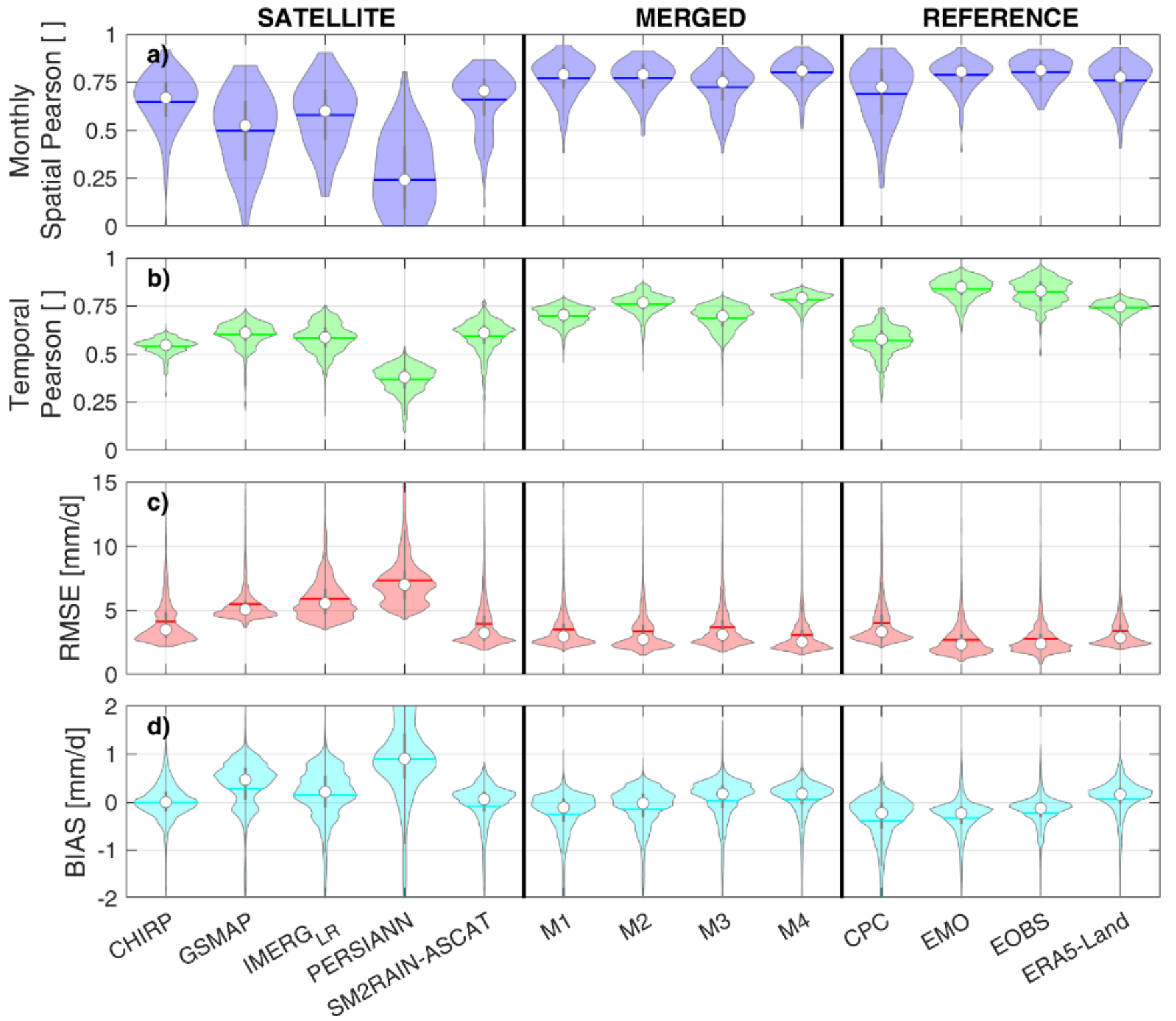

460

**Figure 4: Spatial Pearson's correlation at monthly scale, Temporal Pearson's correlation, BIAS error and RMSE at daily scale for the Ebro River basin against SAIH benchmark (1 km spatial resolution). For each violin, the white dot is the average value, the dark line the median value, and the shape of the violin reflects the data distribution.**

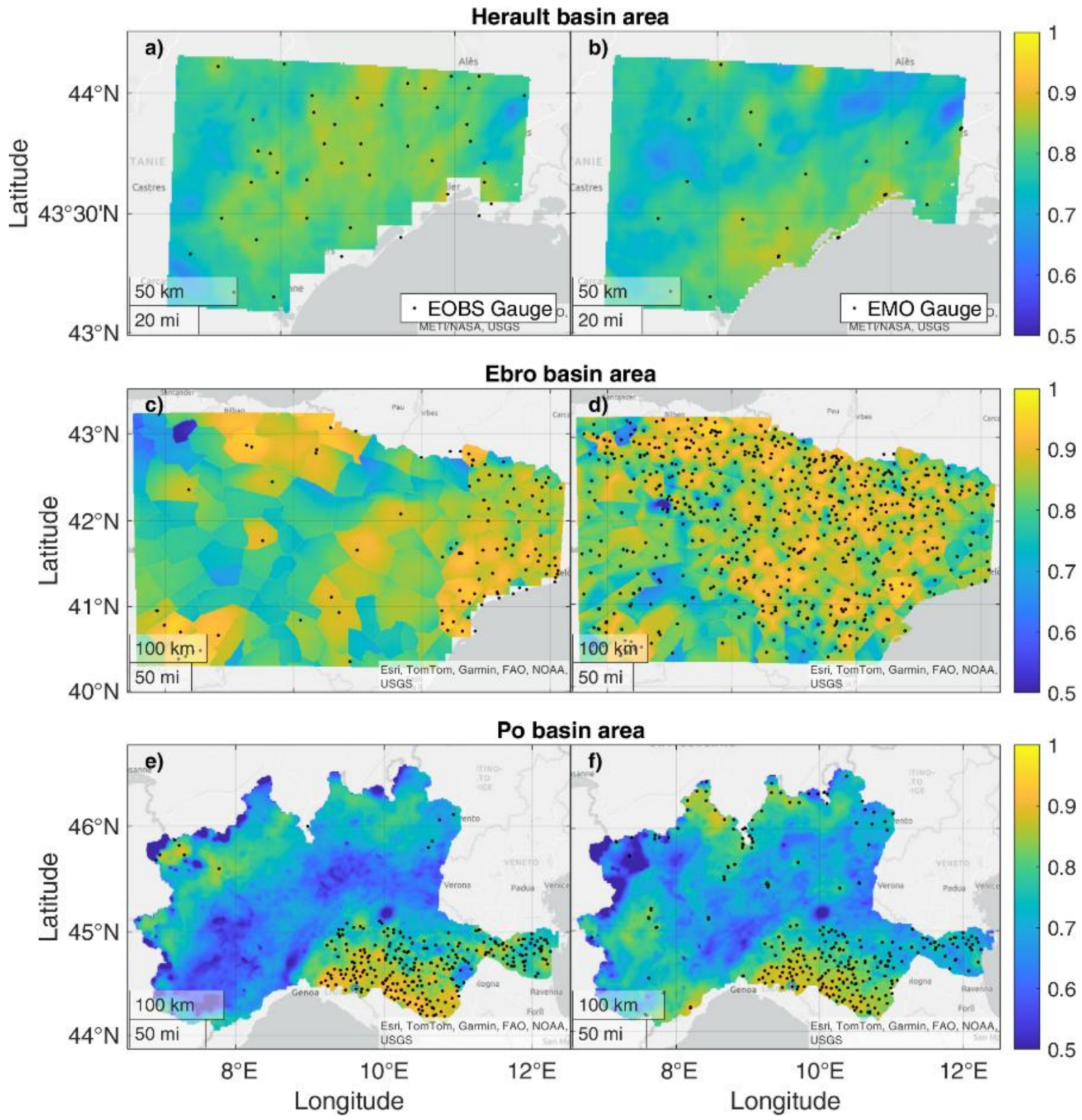

**Figure 5: Pearson's Correlation of E-OBS (a, c, e) and EMO (b, d, f) in Hérault basin area (a, b, COMEPHORE precipitation product), Ebro basin area (c, d, SAIH precipitation product), Po basin area (e, f, MCM precipitation product). Black dots represent the location of the gauge used within E-OBS and EMO, respectively. Only stations with more than 3 years of data between 2007 and 2022 are shown.**

In general, all merged products outperformed the individual satellite datasets, and, in some cases, they surpassed also the reference products. M4 showed the best overall performance, followed by M2, M1, and M3. In the Po River basin, M4 outperformed all the other products across all metrics except for BIAS. The superior performance of M4 can be attributed to the inclusion of ERA5-Land precipitation, which generally outperforms CPC (used in M1 and M2), likely due to its finer spatial resolution and better spatial uniformity (CPC performance is constrained by the locations of the gauges integrated in it). Indeed, the merging of datasets from different approaches clearly benefits precipitation estimation, as confirmed by the spatial distribution of the performance metrics in the Po River basin, shown in Fig. 6-8. Low-performance patterns were observed in all satellite products, including high RMSE in the northwestern portion of the catchment, as well as low Pearson's correlation in specific areas within the Po valley and outside the Italian borders, and the intersecting curved shapes intersecting in the center of the basin within the BIAS results. Since these low performance patterns are similar across all the analyzed products, it is likely that they originated from uncertainty in the MCM benchmark product. Specifically, the RMSE patterns are probably due to noise in the radar measurements due to mountains ground echoes and the low number of gauge stations in that area. The Pearson's correlation patterns could be instead related to the already known absence of gauge outside the Italian

borders within MCM, while the circular low-correlation areas could result from poorly performing, specific gauge stations. Finally, the intersecting curved shapes of the BIAS likely reflect the influence range of radar measurements. Consequently, comparisons in these areas should be treated with caution.

Among the satellite products, as previously mentioned, IMERG-LR and SM2RAIN-ASCAT had the best performances. However, all satellite products were less reliable along the northern, western and southern borders of the Po River basin (Fig. 6a-e), highlighting the challenges of satellite precipitation estimation in complex topographical areas. BIAS maps (Fig. 7) show that all satellite-based precipitation products derived from the TD approach significantly underestimate precipitation in these regions, whereas SM2RAIN-ASCAT tends to overestimate it. This is likely due to the satellite products' limitations in estimating snowfall. Indeed, SM2RAIN-ASCAT does not exhibit underestimation because it measures only liquid precipitation: regions with negative surface temperatures are masked in this product due to the inability to retrieve SM from satellite sensors under frozen conditions. Notwithstanding this, SM2RAIN-ASCAT BIAS is large in those areas because satellite SM estimates in complex topographical regions are of lower quality due to shadowing effects and layover (Ulaby et al., 1981). SM2RAIN-ASCAT product also includes a monthly BIAS correction using ERA5-Land rainfall data (total precipitation - snowfall, Brocca et al. 2019). As a result, its BIAS pattern resembles that of ERA5-Land (e.g., Fig. 7e, 7m), although they are not identical, since ERA5-Land precipitation is analyzed (including snowfall contribution) and in any case the correction is obtained from the climatology of the monthly averages (Brocca et al., 2019). These limitations are inherited by the merged product M3 (Fig. 6h, 7h and 8h), which relies solely on SM2RAIN-ASCAT and IMERG-LR, but are partially mitigated by merging the satellite data with reanalysis data from ERA5-Land (M4) or gauge data from CPC (M1, M2). These merged products exhibit consistently good and uniform performance across all indices in all study areas, though M1 and M3 tend to underestimate precipitation in mountainous regions, while M4 shows a tendency to overestimate, due to ERA5-Land probable overestimation of snowfall. Lastly, it is important to note the varying performance of reference products in the Po Valley. For the gauge-based reference (E-OBS, CPC, EMO) this is likely due to the uneven gauge networks distribution. Nevertheless, all the merged products consistently exhibit strong performance across the region, with M3 outperforming the reference products in the northern part of the Po Valley. This is particularly significant as it demonstrates that satellite-based precipitation products can outperform both reference and models in areas with sparse gauge networks.

The results obtained in the Ebro and Hérault River basins overall corroborate the above findings. For the sake of brevity, they are shown in Appendix B.

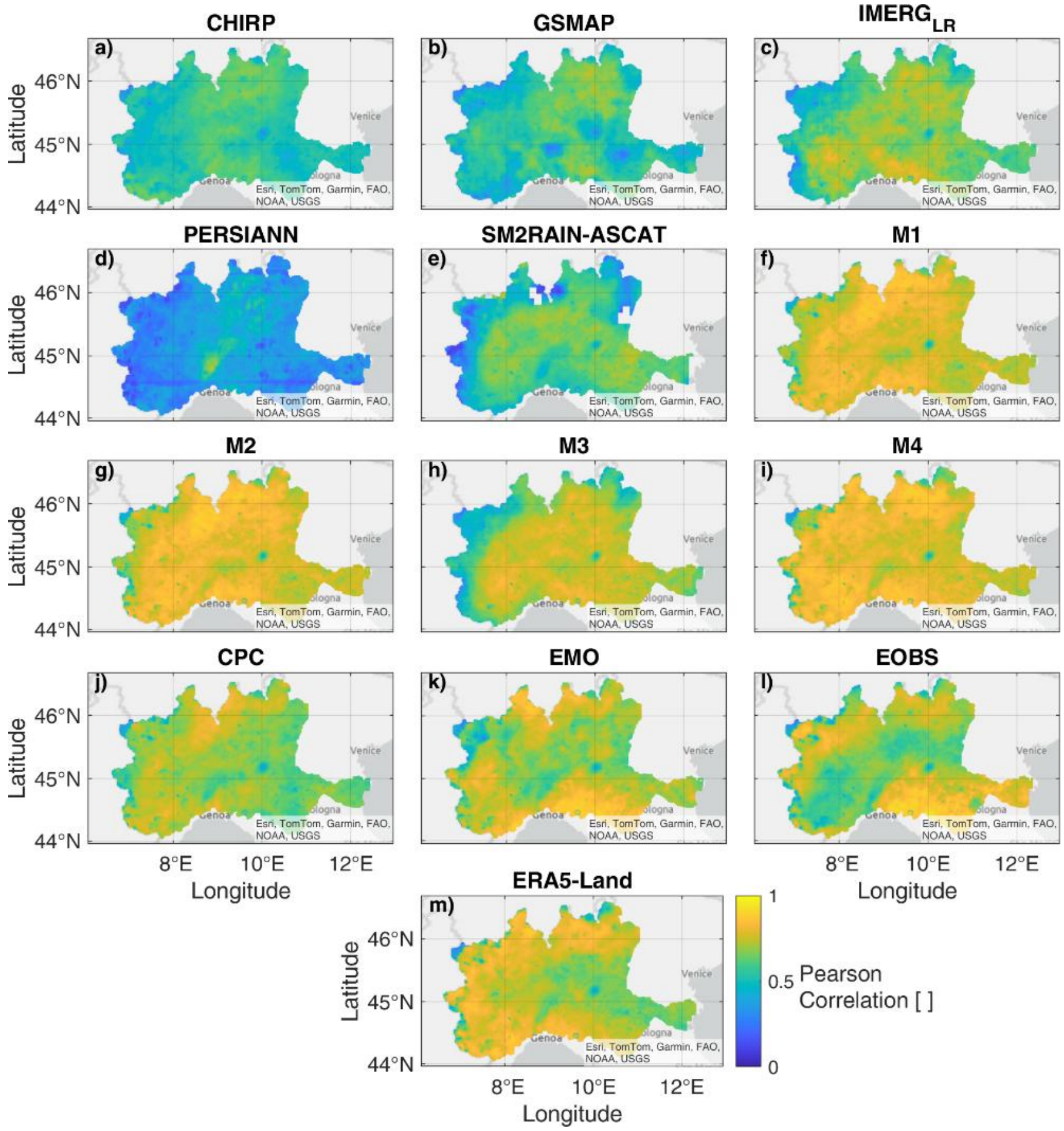

**Figure 6: Daily temporal Pearson's correlation of the selected precipitation datasets against MCM observations for the Po River basin area.**

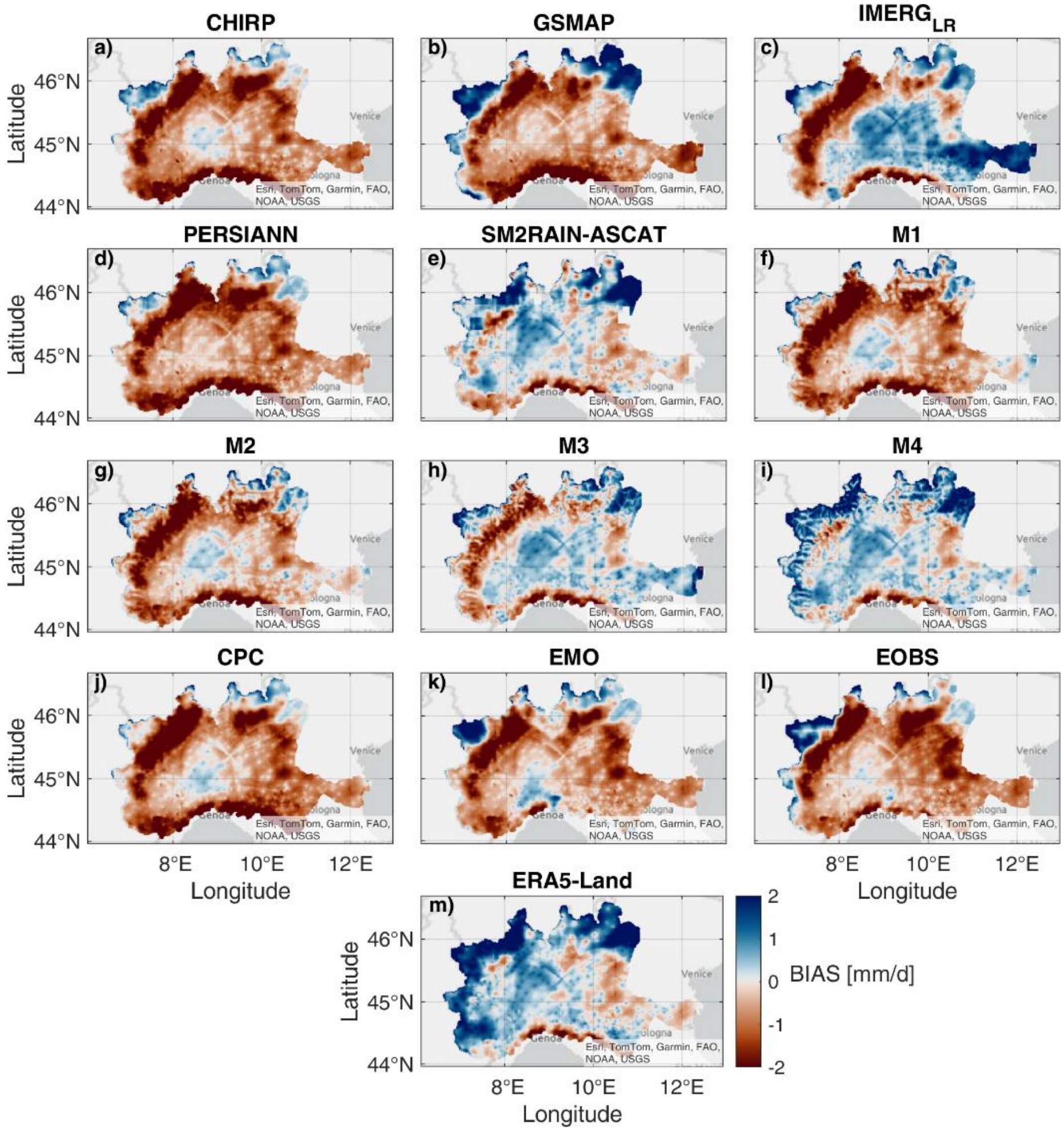

515 **Figure 7: BIAS error of the selected precipitation datasets against MCM observations for the Po River basin area. Blue area means that the precipitation product overestimate precipitation, while brown area means underestimation.**

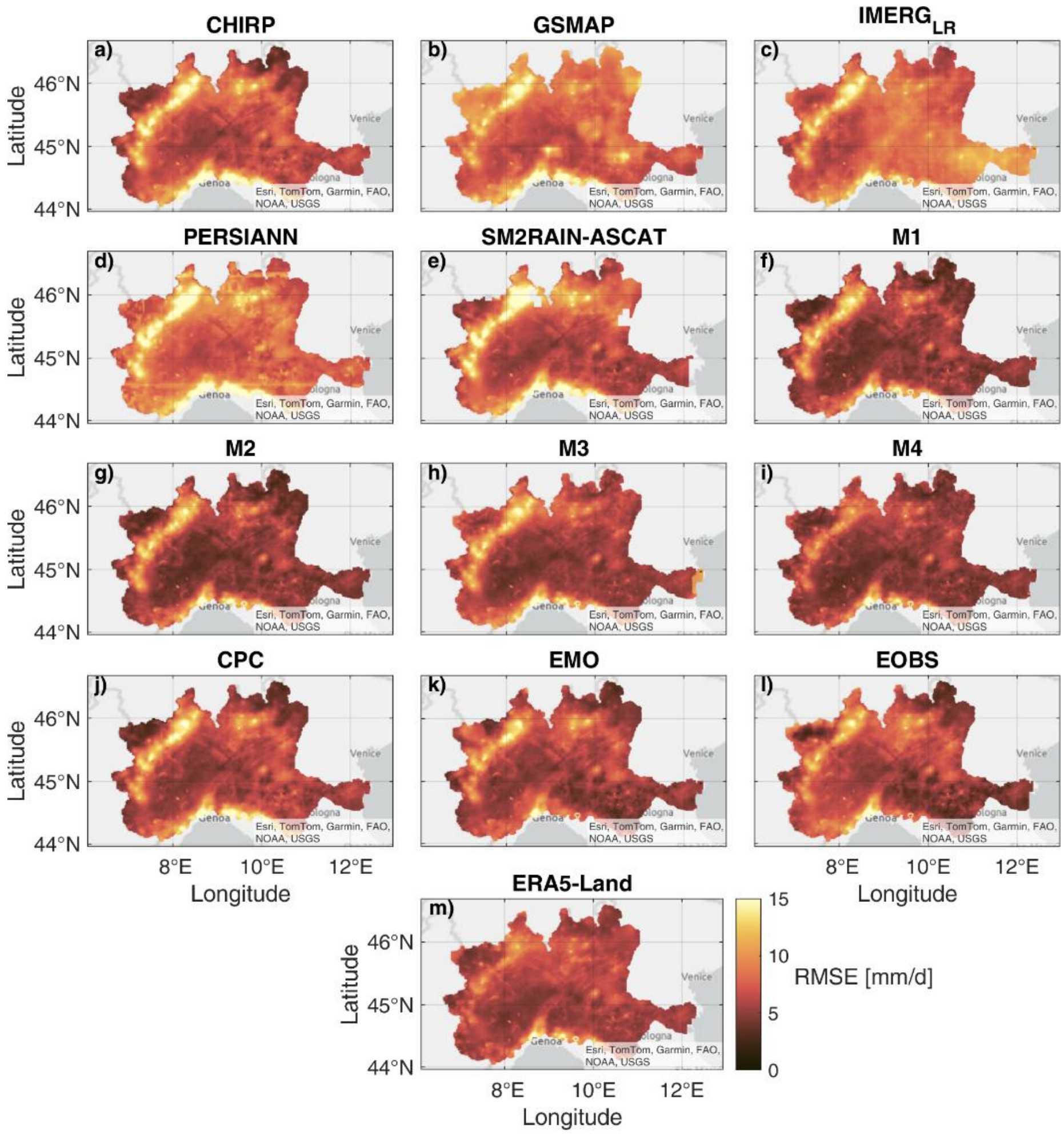

**Figure 8: Root Mean Square Error (RMSE) of the selected precipitation datasets against MCM observations for the Po River basin area.**

### 4.2 Coarse-resolution validation

For the coarse-resolution validation, all products were re-gridded to a regular 0.1-degree grid across the entire study region. The re-gridding was performed using bilinear interpolation when the original spatial resolution of the considered product was greater or equal to the target resolution (~10 km). In cases where the original pixels' dimensions are smaller, spatial aggregation was used. Due to similarities with M4 and lower performances, the coarse-resolution validation of M1 and M2 is not shown. However, M3 is retained due to its independence from any reference measurements. PERSIANN and GSMAP products were also excluded from the comparison because of their low performances in the high-resolution validation. For the coarse-resolution analysis, ERA5-Land was not used as benchmark but kept for comparison due to its inclusion in the M4 product. Similarly, CPC was excluded because of its overall low performance compared with EMO and E-OBS. The validation assessment is conducted across the entire study area, except for the northernmost part of Europe, due to the extensive snow

and ice cover and the low reliability of observations. However, precipitation data for this region are still provided (see the Data Availability section), albeit with caveats due to the lack of validation.

Figures 9 and 10 show the performance of the selected precipitation products over the study area against EMO and E-OBS respectively. It should be noted that CHIRP's assessment is somewhat biased, as its spatial coverage only partially overlaps with the study areas (CHIRP is available between latitude -60 and 60). The results from the high-resolution validation are confirmed here, with the merged satellite products performing very well against both EMO and E-OBS. The merged product based solely on satellite data outperforms all individual satellite products in most indices, except for RMSE, impacted by the high errors from IMERG-LR.

M3 generally performs worse than the reference datasets but merging with ERA5-Land allows notable improvements across all performance indices, with average results on par with those of the reference products. The violin plots of the temporal Pearson's correlation and RMSE between EMO and E-OBS again reveal a double-edge pattern, indicating that the two datasets are similar in at least part of the study area. This is further confirmed by Fig. 11, where a clear overlap between portions of EMO and E-OBS gauge networks is visible. Indeed, EMO benefits from a larger pool of data. However, the availability of certain gridded datasets used (e.g., CarpatClim, Euro4m-APGD, CombiPrecip) fluctuate over time (Thiemig V. et al., 2020), which suggests that the accuracy of EMO may also vary based on data availability.

Performance index maps are displayed in Fig. 12-15, showing Pearson's correlation against EMO and E-OBS (Fig. 12-13), RMSE against EMO (Fig. 14) and BIAS against E-OBS (Fig. 15) for the entire study area. These maps indicate that the northern, central, and western parts of Europe exhibit high Pearson's correlation, low RMSE and near-zero BIAS between EMO and E-OBS due to the overlap of rain gauge stations used as data sources. However, performance declines in the remaining regions. A high degree of agreement between EMO and ERA5-Land is observed in eastern and southern Europe (12g, 14g, 15g), owing to EMO's incorporation of ERA-interim data while a lower correspondence can be noticed between E-OBS and ERA5-Land in these same areas (Fig. 13g). The RMSE and BIAS results against E-OBS are similar to those obtained from EMO and therefore they are not shown here. They are available in Appendix A, for completeness.

Among the satellite products, CHIRP shows good Pearson's correlation over western Spain and northern and central Italy (Fig. 12a, 13a), almost on par with IMERG-LR (Fig. 12b, 13b). IMERG-LR generally outperforms CHIRP in the remaining areas in terms of Pearson's correlation, but CHIRP has lower RMSE and BIAS (Fig. 14a, 15a) than IMERG-LR (Fig. 14b, 15b). Indeed, IMERG-LR exhibits large RMSE and BIAS errors across the entire study area, particularly along coastlines. Notably, IMERG's performance improves beyond 60 degrees of latitude, likely due to an intense masking of the snowy/icy period beyond this latitude (Huffman et al., 2019).

SM2RAIN-ASCAT also performs well in central and western Europe (12c-15c), except in topographically complex areas (due to the above-mentioned issue in the SM estimation) and along coastlines. The low performance and missing data near the coastlines in SM2RAIN-ASCAT are due to an issue in ASCAT SM data (H SAF h119 and h120), specifically due to an erroneous masking. This issue is expected to be resolved in future product versions, which could potentially lead to

improvements in both the SM2RAIN-ASCAT product and the associated merged datasets. Here, missing SM2RAIN-ASCAT data are replaced by IMERG-LR in the M3 merged product, which reduces the drop of the Pearson correlation but causes high RMSE in these areas (Fig. 14d). Adding ERA5-Land within M4 merged product improves performance, as mentioned before, but the effect is different according to the selected benchmark: the performance improves over northern, central and western Europe for both the datasets, with increases in Pearson's correlation and reductions in both RMSE and BIAS. In the eastern region, however, low Pearson's correlation persists against E-OBS, despite the addition of ERA5-Land (Fig. 13e), even though the RMSE decreases (Fig. 14e). In contrast, the comparison with EMO shows more substantial improvements (Fig. 12e), likely due to the strong correspondence between EMO and ERA5-Land in the region. Indeed, the low gauge density in eastern Europe contributes to the uncertainty in this region, as the benchmark datasets lack sufficient rain gauge data for accurate precipitation estimates. The absence of gauge stations in this area limits the reliability of both EMO and E-OBS products, which are based on spatial interpolation techniques (Cornes et al., 2018). As an example, EMO uses data from ERA-Interim, thus explaining the accordance with ERA5-Land in this region. However, the results of the high-resolution validation show that these sources are not always accurate (e.g. Fig. 6m). All the merged satellite datasets combining TD and BU approaches perform well in regions where satellite measurements are reliable (e.g., excluding mountainous areas). This raises important questions about whether the low performances observed in individual satellite products and their merged versions in these areas is due to satellite limitation in estimating precipitation or the inadequacy of the reference products in accurately estimating precipitation patterns in the region.

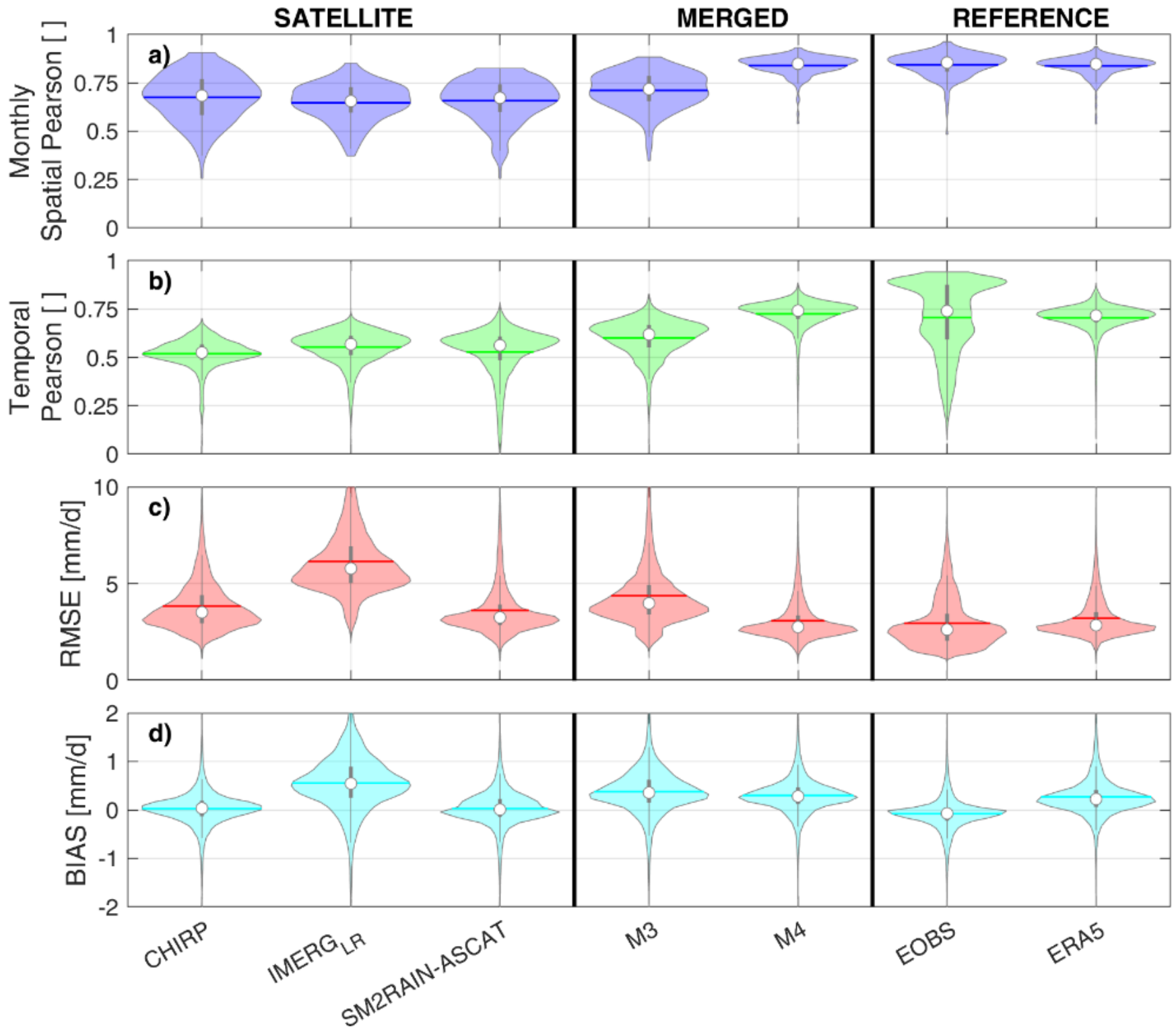

**Figure 9: Spatial Pearson's correlation at monthly scale, Temporal Pearson's correlation, BIAS error and RMSE at daily scale for the full study area against EMO benchmark (10 km spatial sampling). For each violin, the white dot is the average value, the dark line the median value, and the shape of the violin reflects the data distribution.**

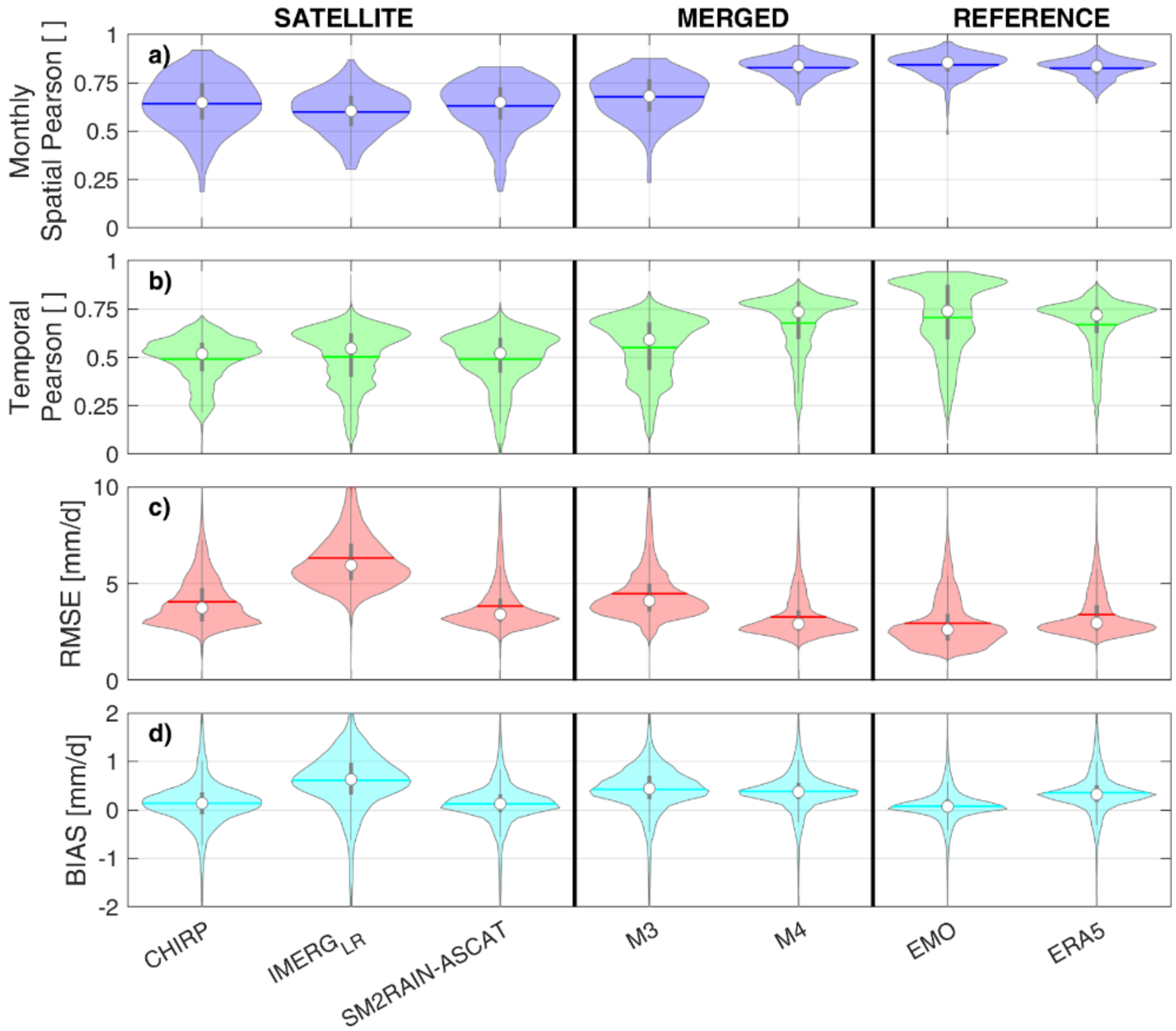

**Figure 10: Spatial Pearson's correlation at monthly scale, Temporal Pearson's correlation, BIAS error and RMSE at daily scale for the full study area against E-OBS benchmark (10 km spatial sampling). For each violin, the white dot is the average value, the dark line the median value, and the shape of the violin reflects the data distribution.**

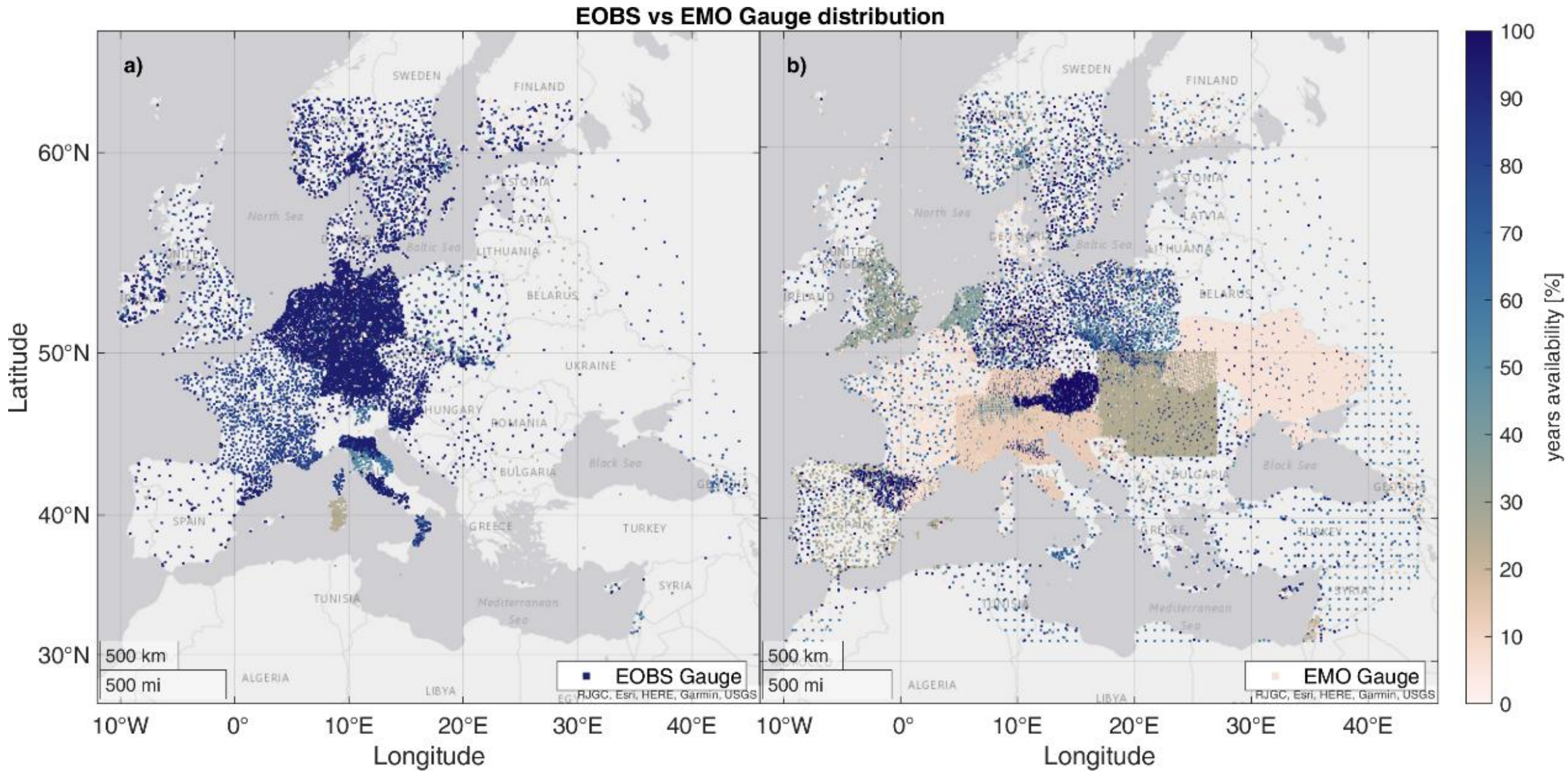

**Figure 11: Pearson's Correlation of E-OBS against EMO in the full study area. Black dots represent the location of the gauge used within E-OBS (a) and EMO (b) datasets, respectively. The years availability is obtained by calculating the number of years between 2007-2022 for which data from each pixel were available.**

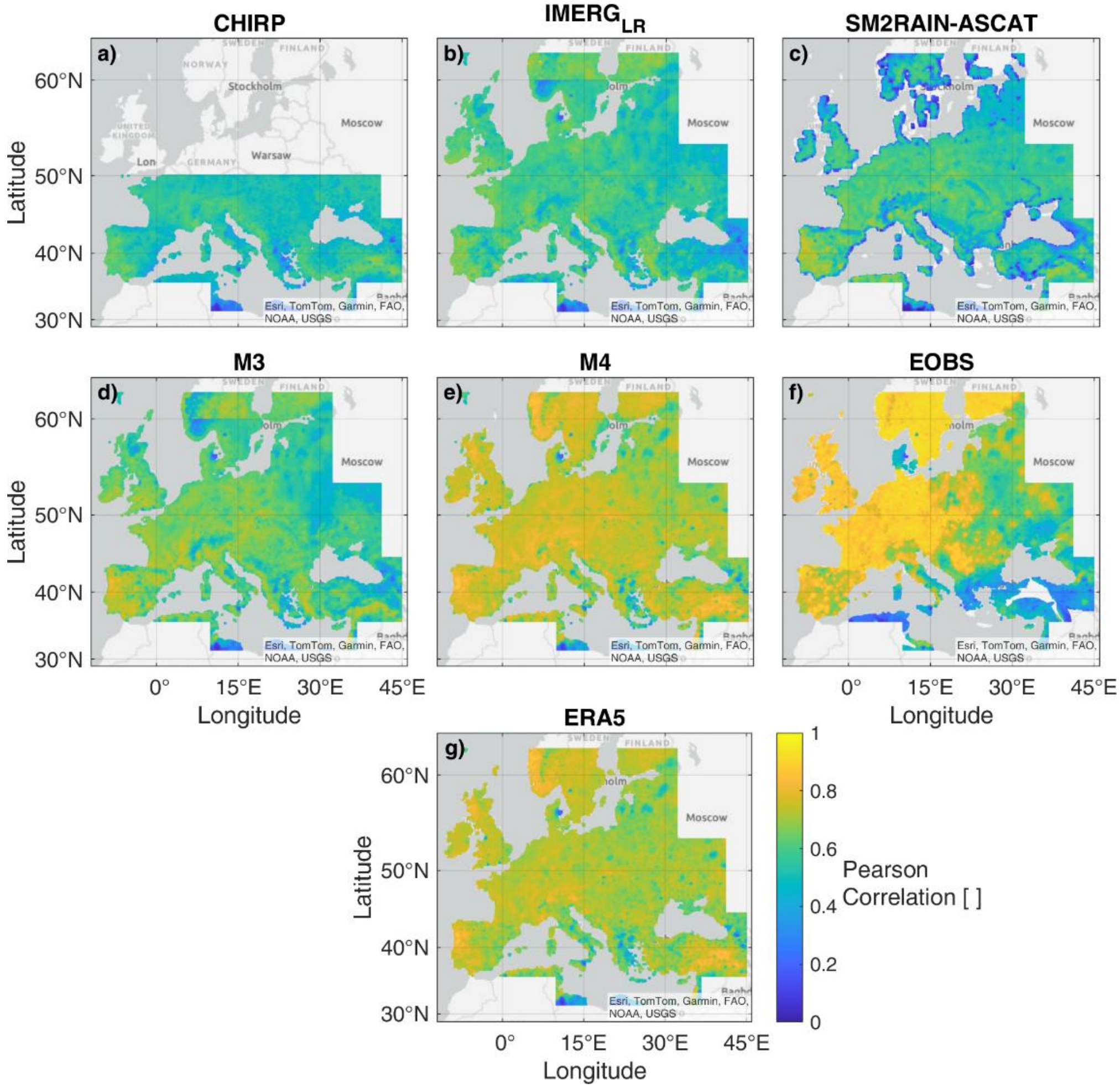

**Figure 12: Daily temporal Pearson's correlation of the selected precipitation datasets against EMO observations for the full study area. All the datasets are aggregated at 10 km spatial resolution.**

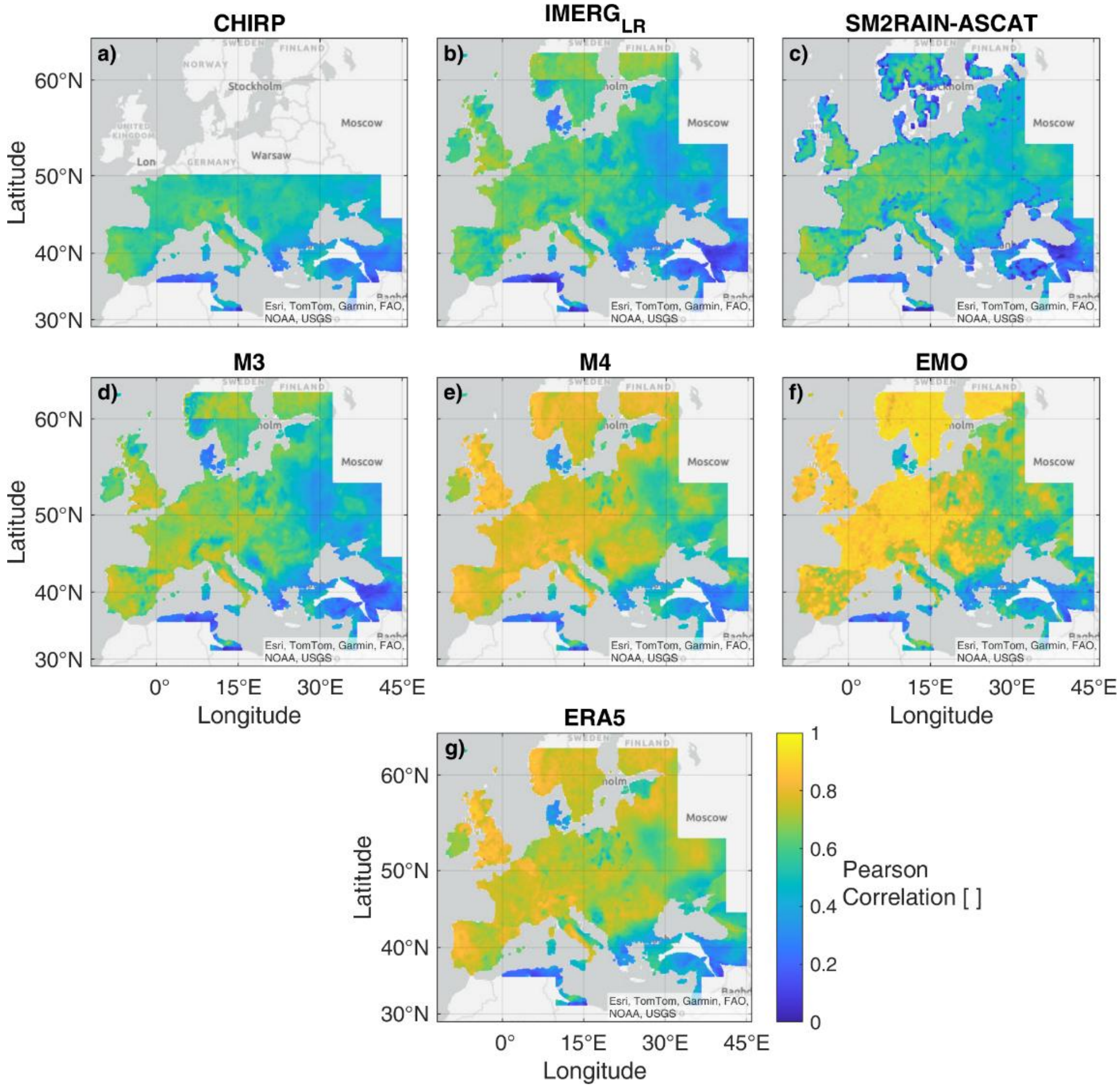

**Figure 13: Daily temporal Pearson's correlation of the selected precipitation datasets against E-OBS observations for the full study area. All the datasets are aggregated at 10 km spatial resolution.**

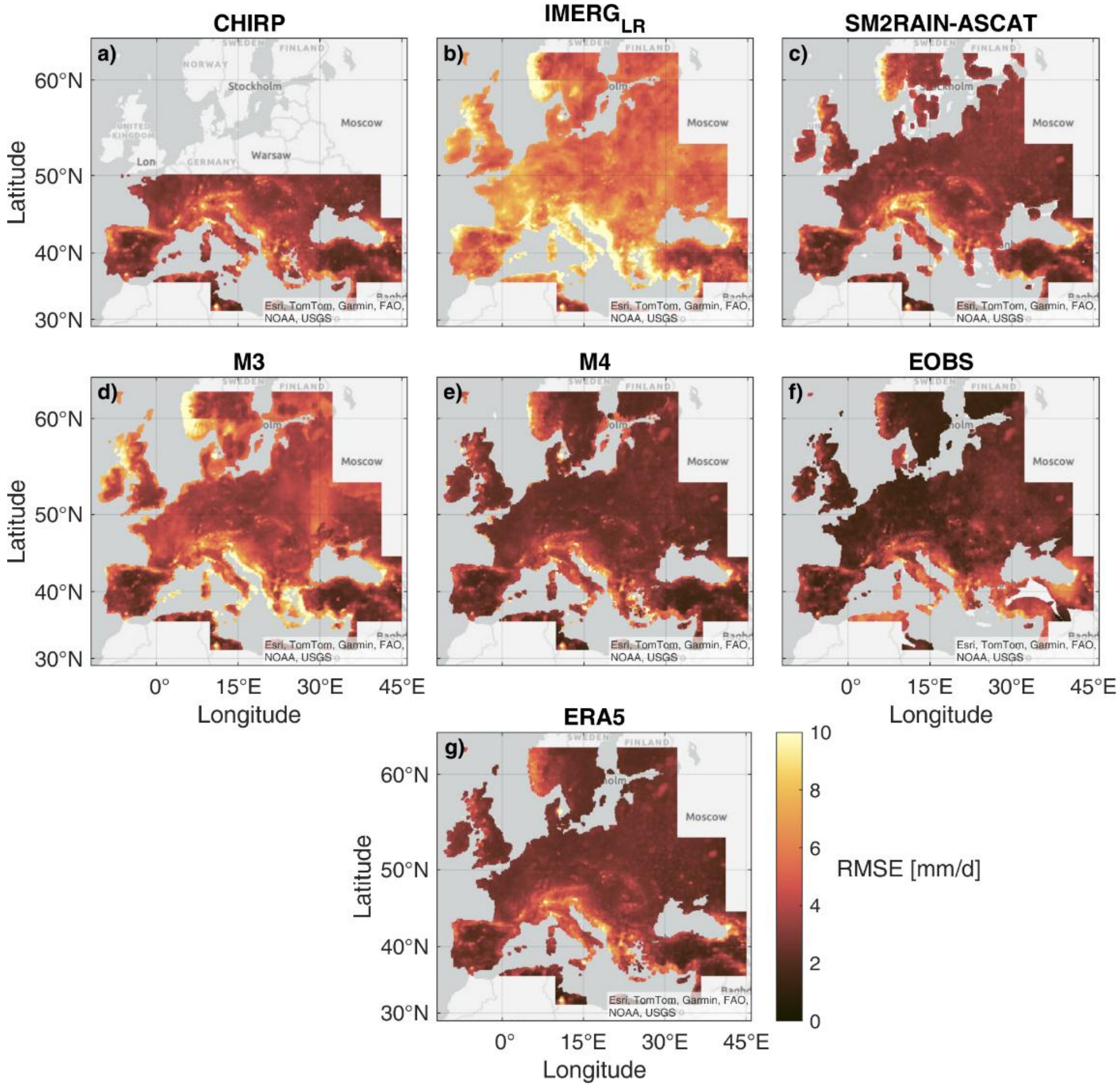

**Figure 14: Root Mean Square Error (RMSE) of the selected precipitation datasets against EMO observations for the full study area. All the datasets are aggregated at 10 km spatial resolution.**

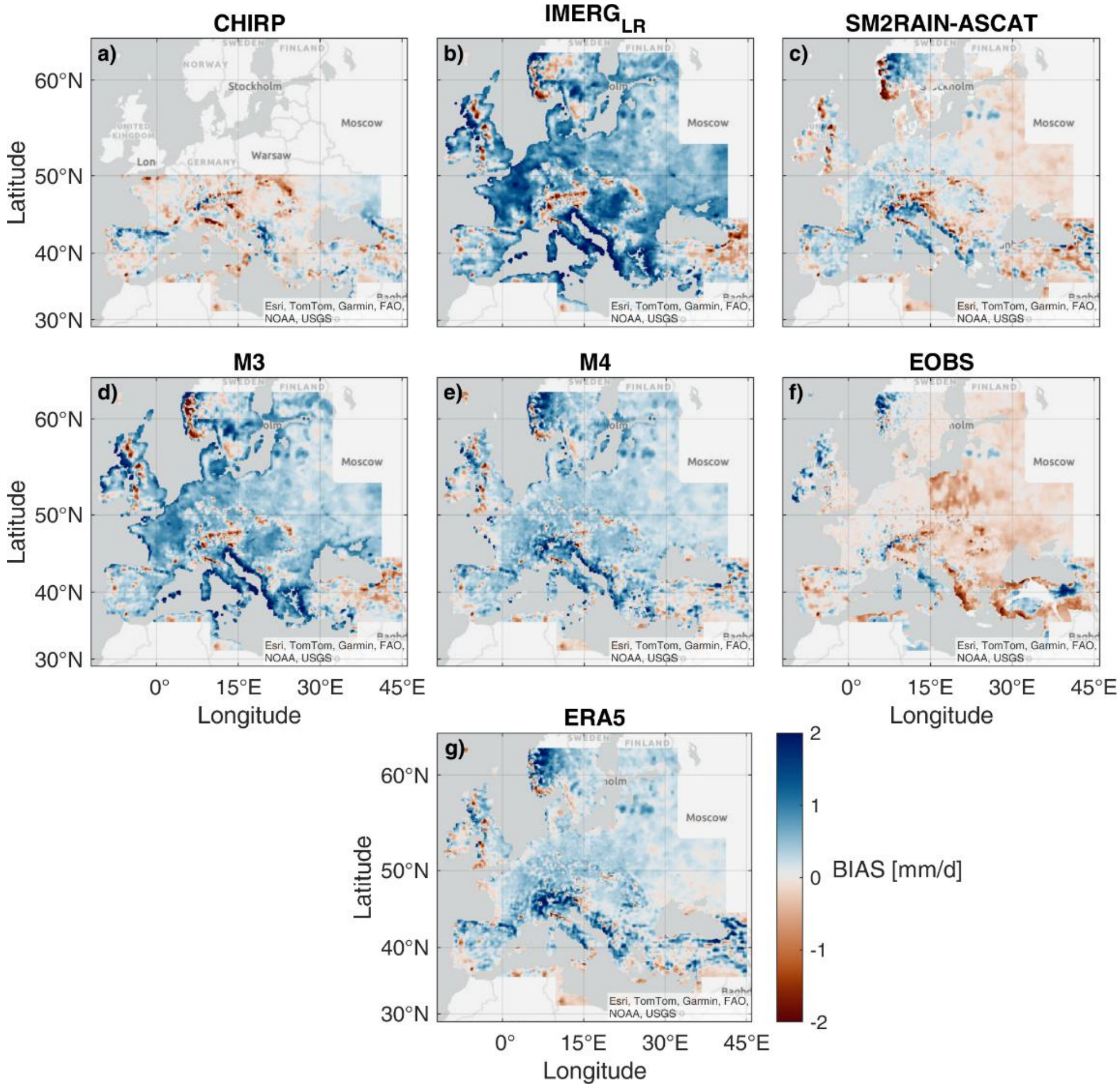

**Figure 15: BIAS error of the selected precipitation datasets against EMO observations for the full study area. Blue area means that the precipitation product overestimate precipitation, while brown area means underestimation. All the datasets are aggregated at 10 km spatial resolution.**

### 4.3 Discussion

The validation analysis assessed the performance of the four integrated precipitation products against coarse and high-resolution observed data. Among the four products, the best performing ones were the configurations M3 and M4, based respectively only on satellite data and on satellite data plus reanalysis. This is probably due to the native coarse resolution of the CPC dataset used in configuration M1 and M2, as well its relatively low number of included rain gauges. M3 evaluation indicates that this configuration may provide valuable precipitation information for those areas where rain gauge networks are less dense. However, it was demonstrated that M4 configuration is generally the most reliable, in particular for those areas in which satellite data are known to be less performing, e.g. mountainous environment. Therefore, the M4 configuration is selected as the best performing and it is named HYPER-P (Figure 16).

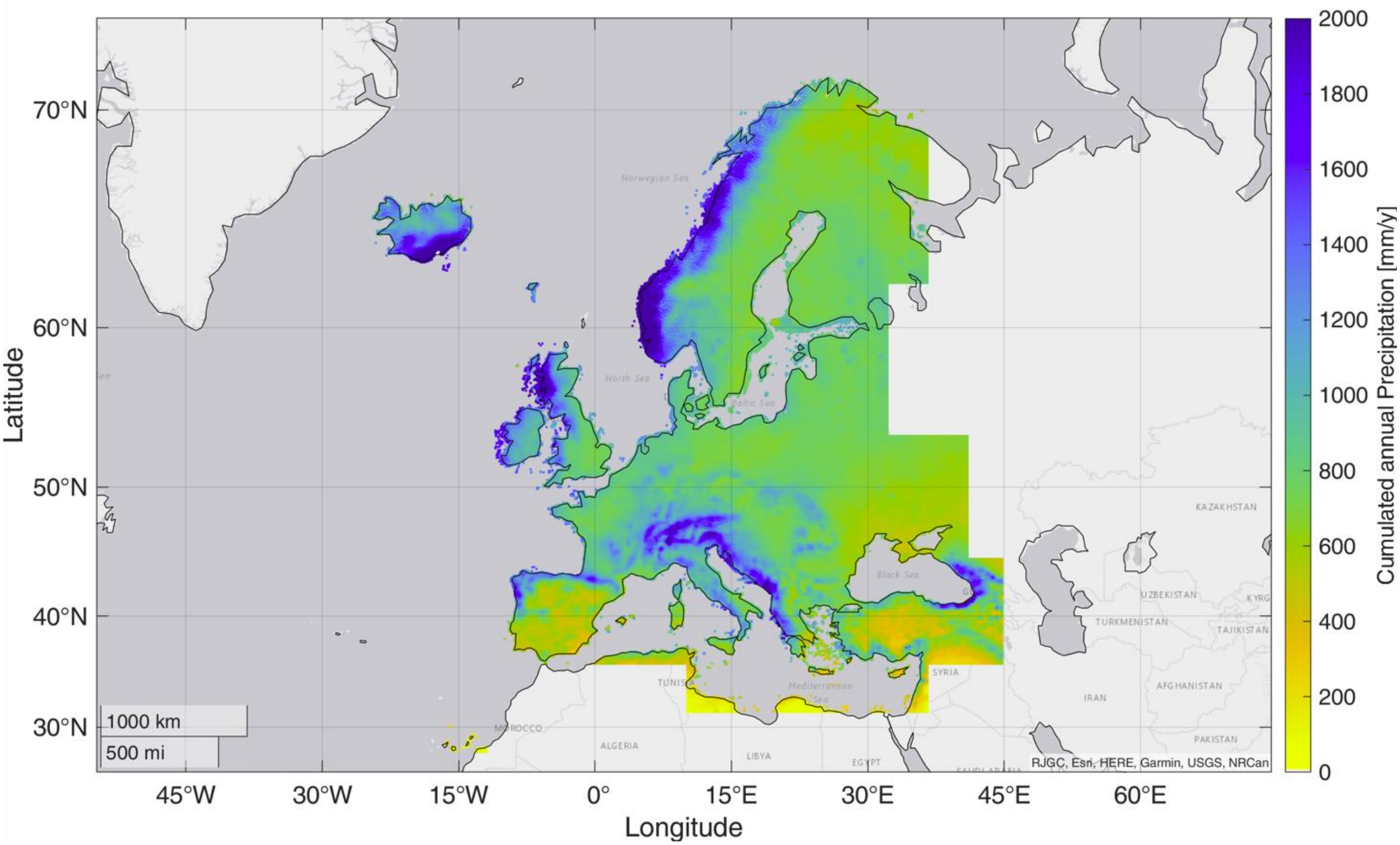

**Figure 16: Cumulated annual precipitation map of HYPER-P precipitation product, obtained by downscaling and merging precipitation product from reanalysis, satellite TD and satellite BU: specifically, ERA5-Land, IMERG Late Run and SM2RAIN-ASCAT.**

This gridded precipitation product integrates multiple observational sources, leveraging both satellite-based retrievals and reanalysis data with a high spatial resolution (1 km). It has been developed to enhance spatial consistency and temporal coverage of the parent product, improving precipitation estimation compared to individual datasets. However, some limitations remain. First, the dataset is available only from 2007 onward, according to ASCAT (and SM2RAIN-ASCAT) data availability, limiting its use in long-term climate studies. Second, satellite-based precipitation estimates exhibit lower performance over complex topography, particularly in mountainous regions where both gauge-based and satellite-derived products tend to be less reliable due to orographic effects and snow-related biases (Girotto et al., 2024). These limitations are expected to propagate into the merged product, together with potential underestimation from satellite-derived estimates. Similarly, also ERA5-Land limitations could propagate to the new product (e.g. higher error in case of convective precipitation, Lavers et al., 2022). Finally, the original coarse resolution of the parent datasets may generate issues on localized convective precipitation events, since the downscaling procedure is based on monthly climatologies. However, using weights derived from TC reduces dependency on the parent products in areas where they perform poorly, as the relative weight decreases, thereby minimizing propagated issues.

Despite these constraints, the product is expected to provide valuable insights into precipitation patterns that will be useful in multiple research fields. This is demonstrated by the use of the developed precipitation products in multiple recent studies:

- In Brocca et al., 2024, the integrated products were used in the context of the Digital Twin Earth and validated against a hydrological model over the Po Valley, obtaining Kling-Gupta Efficiency values higher than the observed products;
- M1 configuration was used in Pellet et al., 2024 in a simple hydrological model of the Ebro River, resulting in good performance, closer to those obtained with an assimilation scheme;
- M2 configuration was used in Peiro et al., 2024 within a Random Forest machine learning model for predicting landslide susceptibility, achieving high predictive accuracy, effectively identifying landslide-prone areas in Italy;
- Sivelle et al. (under review) uses M2 configuration when testing multiple Precipitation and Soil Moisture products as hydrological model input for 5 catchments in Spain, France, Italy, Tunisia and Algeria, obtaining better performance than single satellite products, and showing positive improvements related to the use of the downscaling procedure;
- M2 configuration was used also in Camici et al. (under review) to create a Water Resources Management database for the Po River, with promising results;
- M2 configuration was also used by Dari et al., 2024, within the development of a novel approach for estimating groundwater recharge, obtaining performance similar to those generated by using gauge data;

In some cases, the integrated products did not provide additional information, as in the case of Al Khoury et al., 2024, where the M2 product was used in a small karst catchment in the French Pyrenees, focusing on data-scarce mountainous regions, for hydrological modelling. Here, the use of the merged product did not improve the model performances, confirming the limitation of the dataset for mountainous regions and local convective events. Notwithstanding this, the large use of M1 and M2 demonstrates both their usefulness and the need of the scientific community for high spatial resolution precipitation products. M3 and M4, instead, are still to be tested, since they were only recently developed in the framework of the 4dHydro project.

## 5 Conclusions

In this study, a new precipitation product named HYPER-P, characterized by high spatial resolution and accuracy, is derived for Europe and mediterranean basin from downscaling and merging multiple data sources, including remote sensing products, reanalysis datasets, and gridded in situ observations, in the period 2007 to 2022. For this purpose, multiple precipitation products derived from gauges, radars, reanalysis and satellite observations are mutually compared. A total of twelve different precipitation products – five from satellites, six from reference products and one from reanalysis – were collected for the study area and re-gridded to resolutions of 1 km and 10 km. For each type of precipitation product, datasets characterized by high performance metrics, broad spatial coverage and low latency availability were downscaled and merged to obtain enhanced precipitation products that leverage the complementary strengths of each estimation approach. Specifically, four merged products were developed: M1, satellite TD approach (IMERG-LR) + gauge (CPC); M2, satellite TD approach (IMERG-LR)

+ satellite BU approach (SM2RAIN-ASCAT) + gauge (CPC); M3, satellite TD approach (IMERG-LR) + satellite BU approach (SM2RAIN-ASCAT); and M4, satellite TD approach (IMERG-LR) + satellite BU approach (SM2RAIN-ASCAT) + reanalysis (ERA5-Land). The performance of the merged datasets, individual satellite products and reference datasets was evaluated through two validation analyses conducted at high and coarse spatial resolution, respectively. The high-resolution analysis was performed across three regions in the Mediterranean Basin with dense gauge/radar networks: the Po River Basin, the Hérault River Basin and the Ebro River Basin. This analysis allowed the selection of the best performing products and understanding the mutual limitations of the datasets, such as the low accuracy of satellite products over the mountainous regions and the performance drop of reference datasets in areas with low gauge density. A selection of products was subsequently assessed over most of Europe and Mediterranean basin within the coarse-resolution validation, where all the data was aggregated to a 10 km resolution and then compared against EMO and E-OBS reference datasets.

Satellite data demonstrated a generally strong capability in estimating precipitation, and the combination of BU and TD approaches effectively leverages the strengths of both SM2RAIN-ASCAT and IMERG-LR. This merged product estimate precipitation with high reliability across most of the analyzed areas, even outperforming datasets based on in-situ and reanalysis data in regions with low gauge density. The inclusion of ERA5-Land further enhances these results, particularly improving precipitation accuracy in topographically complex regions where satellite data alone often struggles to achieve good results. However, evaluating precipitation products by using traditional techniques in regions with scarce observed data, such as eastern Europe, is not trivial. In these areas, the merged product obtained from the combination of SM2RAIN-ASCAT, IMERG-LR and ERA5-Land seems to be the best performing against EMO and E-OBS benchmarks, but the low reliability of these reference datasets due to the unavailability of gauge measurements raises concerns about the robustness of the findings. Hence, the merged product was selected as HYPER-P product and, along with satellite-only merged product (SM2RAIN-ASCAT and IMERG-LR), will be undertaken a hydrological validation within the ESA 4DHydro project to further assess the optimal precipitation dataset through the capability of reproducing observed discharge. The results of this ongoing project, together with the findings of this analysis, will enable the scientific community to further advance its understanding of available precipitation products, particularly in terms of their respective strengths and weaknesses. This knowledge will potentially contribute to the development of a global, high-resolution precipitation product with short latency, which integrates and complements the various existing datasets. Future improvements will focus on refining the downscaling methodology by incorporating higher-resolution datasets, such as Sentinel-1-derived soil moisture, to enhance spatial detail and accuracy, particularly in regions where traditional precipitation estimates remain uncertain. Efforts will also aim to extend the dataset both spatially and temporally, especially in areas with sparse rain gauge coverage.

**Data availability:**

The merged products analyzed in this study were developed within the ESA projects 4DMED and 4DHydro. They are available online at:

M1: IMERG-LR+CPC. https://stac.eurac.edu/browser/#/collections/rainfall_all_domain/ and https://zenodo.org/records/15025397, available for the Mediterranean basin for the period 2000-2022 (Filippucci et al, 2023a).

M2: SM2RAIN-ASCAT+IMERG-LR+CPC. https://zenodo.org/records/10402392, available for the Mediterranean basin for the period 2015-2022 (Filippucci et al, 2023b).

M3: SM2RAIN-ASCAT+IMERG-LR. https://4dhydro.eu/catalog/ (path Products => WP1 Products => 4DHYDRO precipitation product: SM2RAIN+GPM) and https://zenodo.org/records/15025462 (Europe), available for the entire Europe and the Tugela Basin (Africa) from 2007-2022 (Filippucci et al., 2024a)

M4: SM2RAIN-ASCAT+IMERG-LR+ERA5-Land. https://4dhydro.eu/catalog/ (path Products => WP1 Products => 4DHYDRO precipitation product: ERA5+SM2RAIN+GPM) or https://zenodo.org/records/15025514 (Europe), available for the entire Europe and the Tugela Basin (Africa) from 2007-2022 (Filippucci et al., 2024b).

## Appendices

### A Merging weights distribution

The weights used for dataset merging, obtained from the application of the TC, are shown here. Figure A1 presents the weights for configuration M1, where CPC is merged with IMERG-LR. As expected, CPC has the highest weights across most of the northern Mediterranean basin due to the presence of dense meteorological gauge networks. The weights used in configuration M2 are shown in Fig. A2. Here, the results differ slightly: CPC remains the most used product in the northern part of the basin, but IMERG-LR and SM2RAIN-ASCAT also contribute significantly to many areas. Specifically, IMERG-LR is selected over topographically complex regions, where ASCAT SM retrievals (and therefore SM2RAIN-ASCAT rainfall estimations) are less reliable. Figure A3 illustrates the weights for configuration M3, where only SM2RAIN-ASCAT and IMERG-LR are used. In this case, IMERG-LR is the dominant product at higher latitudes, where frozen soil conditions often hinder soil moisture (SM) retrieval and, consequently, rainfall estimation from space. Conversely, SM2RAIN-ASCAT has greater weight in the southern areas. Finally, Figure A4 shows the weights for configuration M4. Here, ERA5-Land has the greatest weight across most of Europe due to the limitations of satellite products in frozen regions. However, in African regions and parts of Eastern Europe, the weights of the satellite datasets increase, likely due to the scarcity of observational data in these areas. Over most of the coastal areas, ASCAT SM data are not available or less accurate, therefore there the SM2RAIN-ASCAT dataset is not used in any configuration.

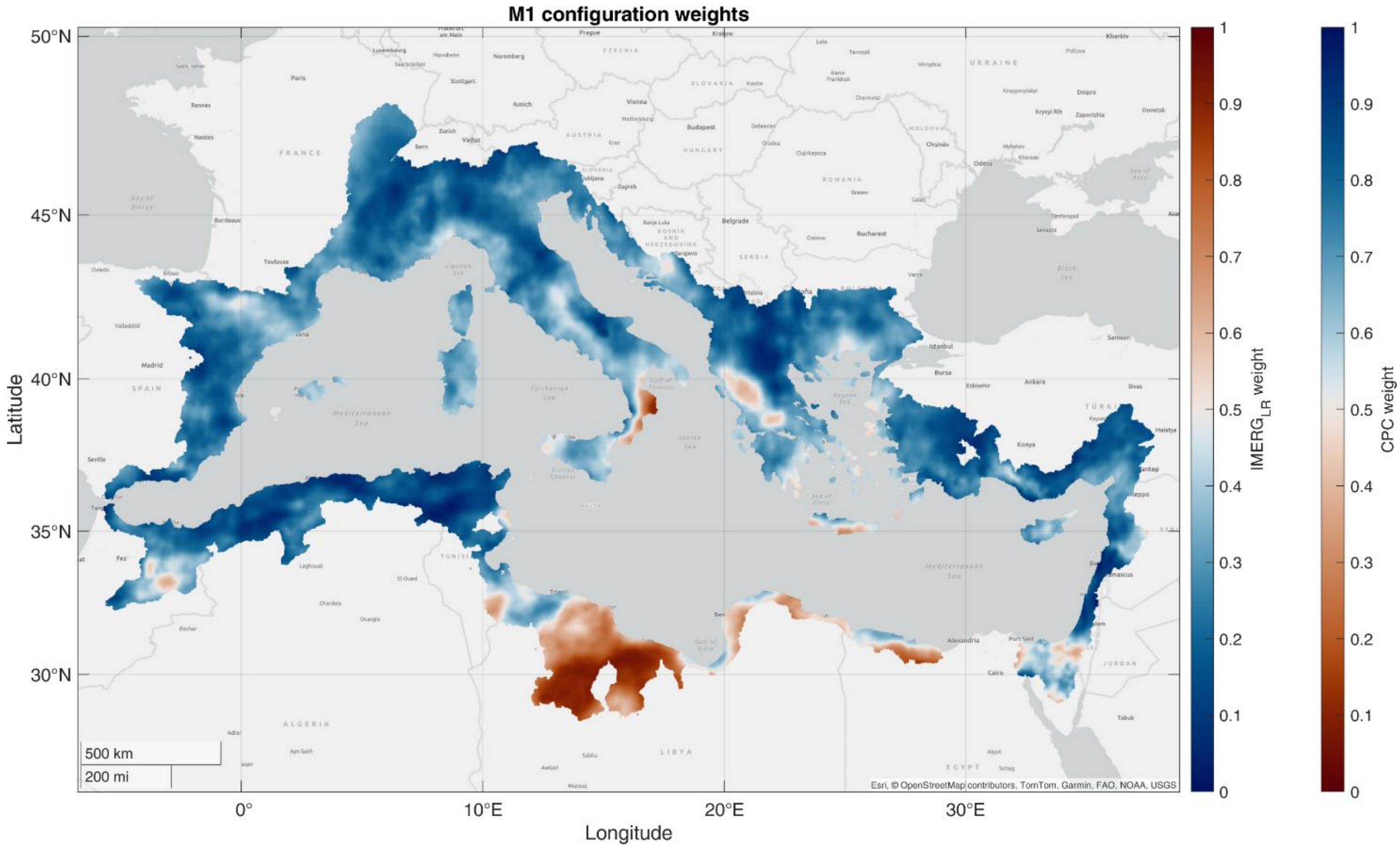

**Figure A1: Weights distribution for configuration M1: areas where CPC has the greater weight are highlighted in blue while those where IMERG-LR is prevalently used are highlighted in brown.**

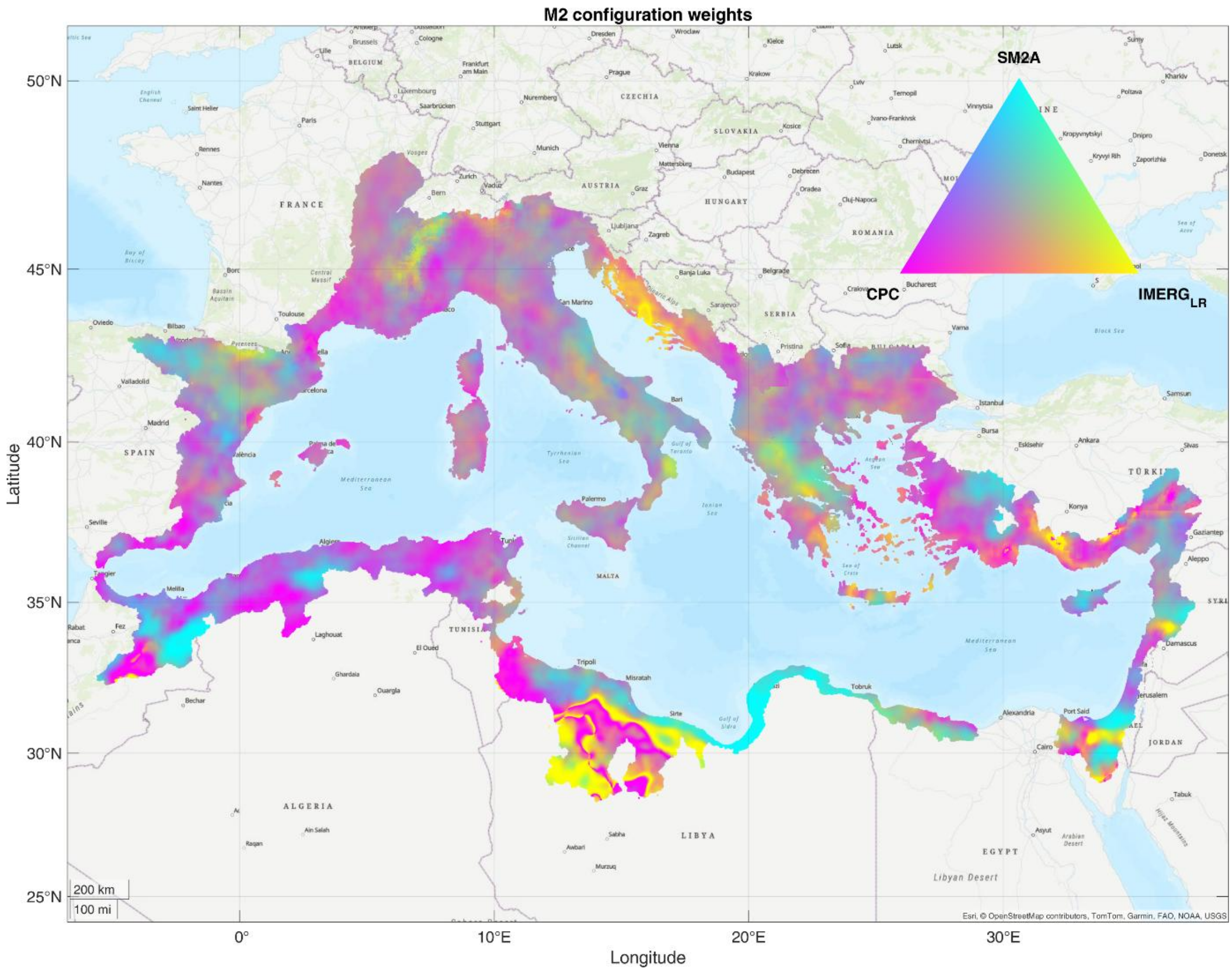

**Figure A2**

**Weights distribution for configuration M2: areas colored in red, green and blue highlight the pixels where CPC, IMERG-LR and SM2RAIN-ASCAT have the greater weight, respectively.**

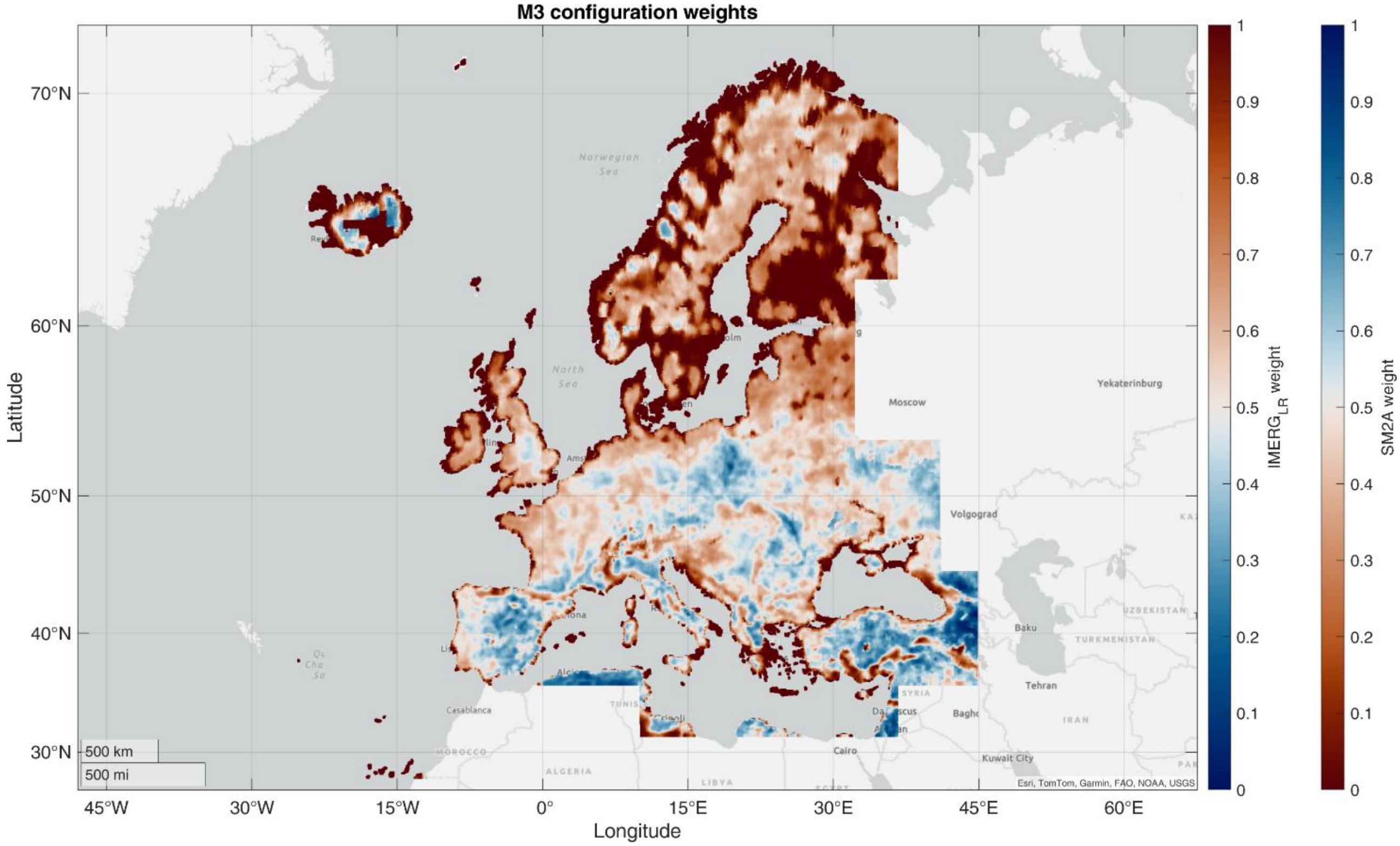

**Figure A3: Weights distribution for configuration M3: areas where SM2RAIN-ASCAT has the greater weight are highlighted in blue while those where IMERG-LR is prevalently used are highlighted in brown.**

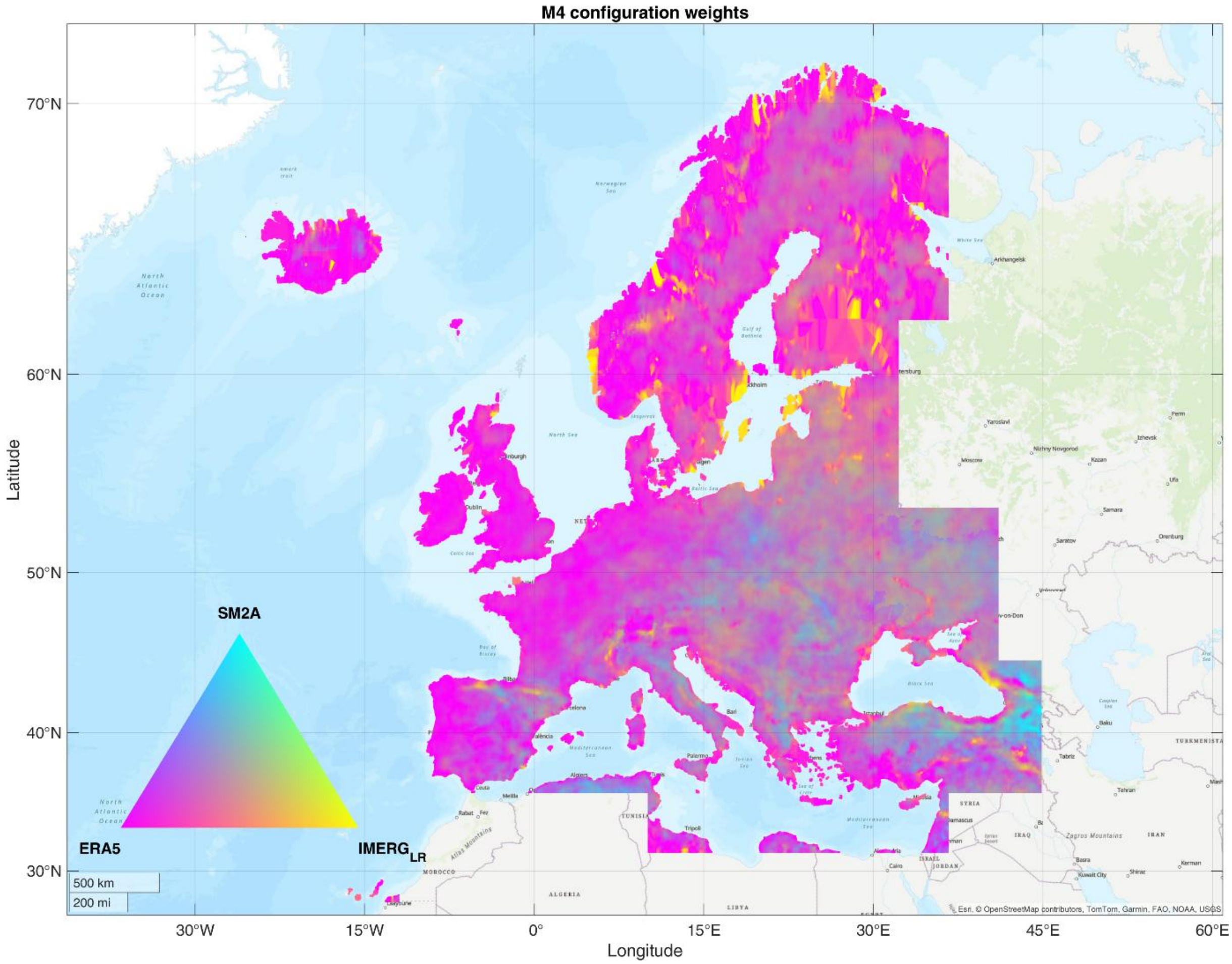

**Figure A4 Weights distribution for configuration M4: areas colored in red, green and blue highlight the pixels where ERA5-Land, IMERG-LR and SM2RAIN-ASCAT have the greater weight, respectively.**

## B Performance indices at high and coarse resolution

Here, the daily Pearson's correlation (Fig. B1, B2), RMSE (Fig. B3, B4) and BIAS (Fig. B5, B6) maps of the high spatial resolution analyzed products against the local high spatial resolution precipitation benchmarks are shown for Ebro River basin (Fig. B1, B3, B5) and the Hérault River basin (Fig. B2, B4, B6). The RMSE (Fig. B7) and BIAS (Fig. B8) performance of the coarse spatial resolution against E-OBS for the full study area are also shown.

resolution against E-OBS for the full study area are also shown.

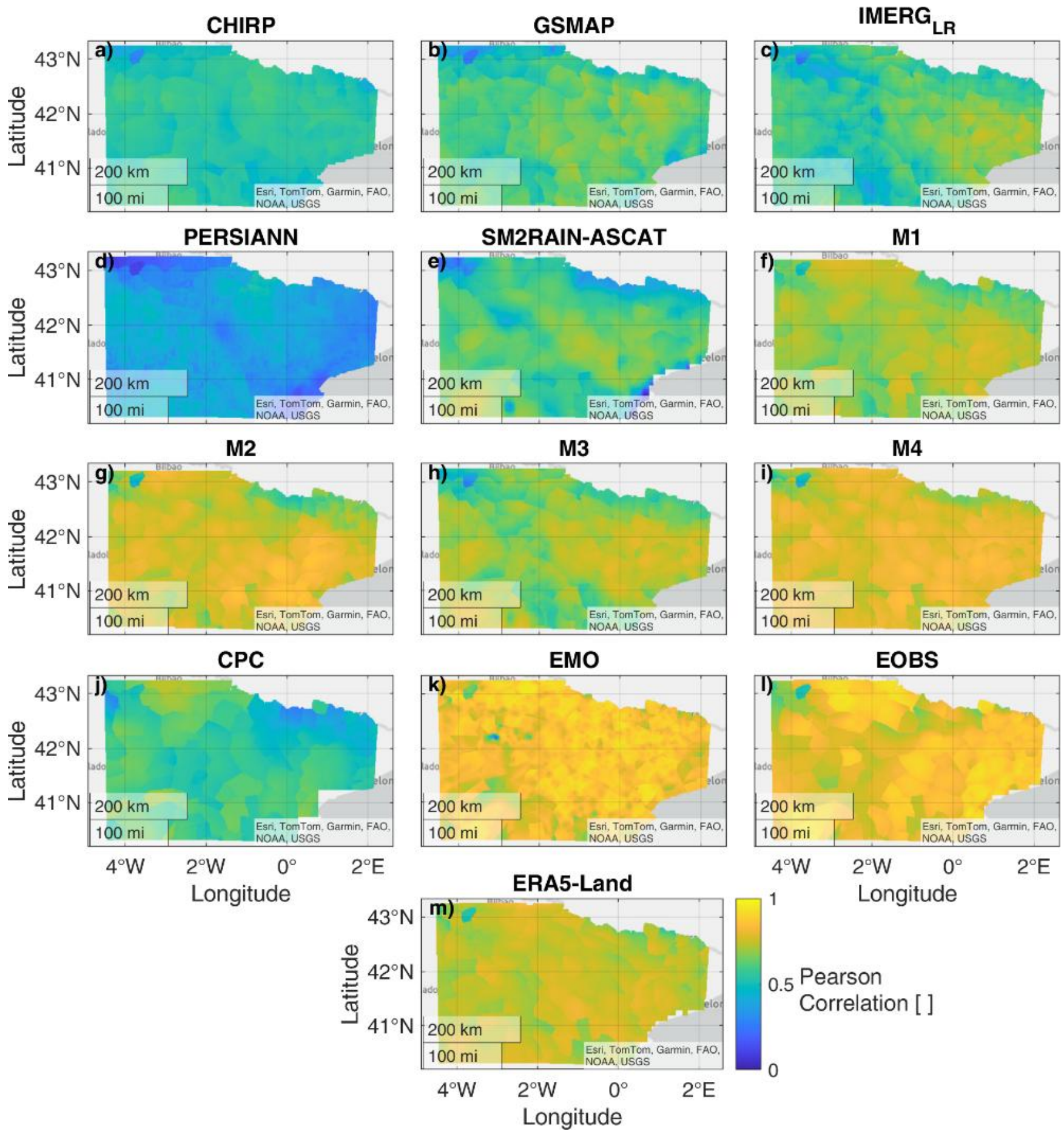

**Figure B1: Daily temporal Pearson's correlation of the selected precipitation datasets against SAIH observations for the Ebro River basin area.**

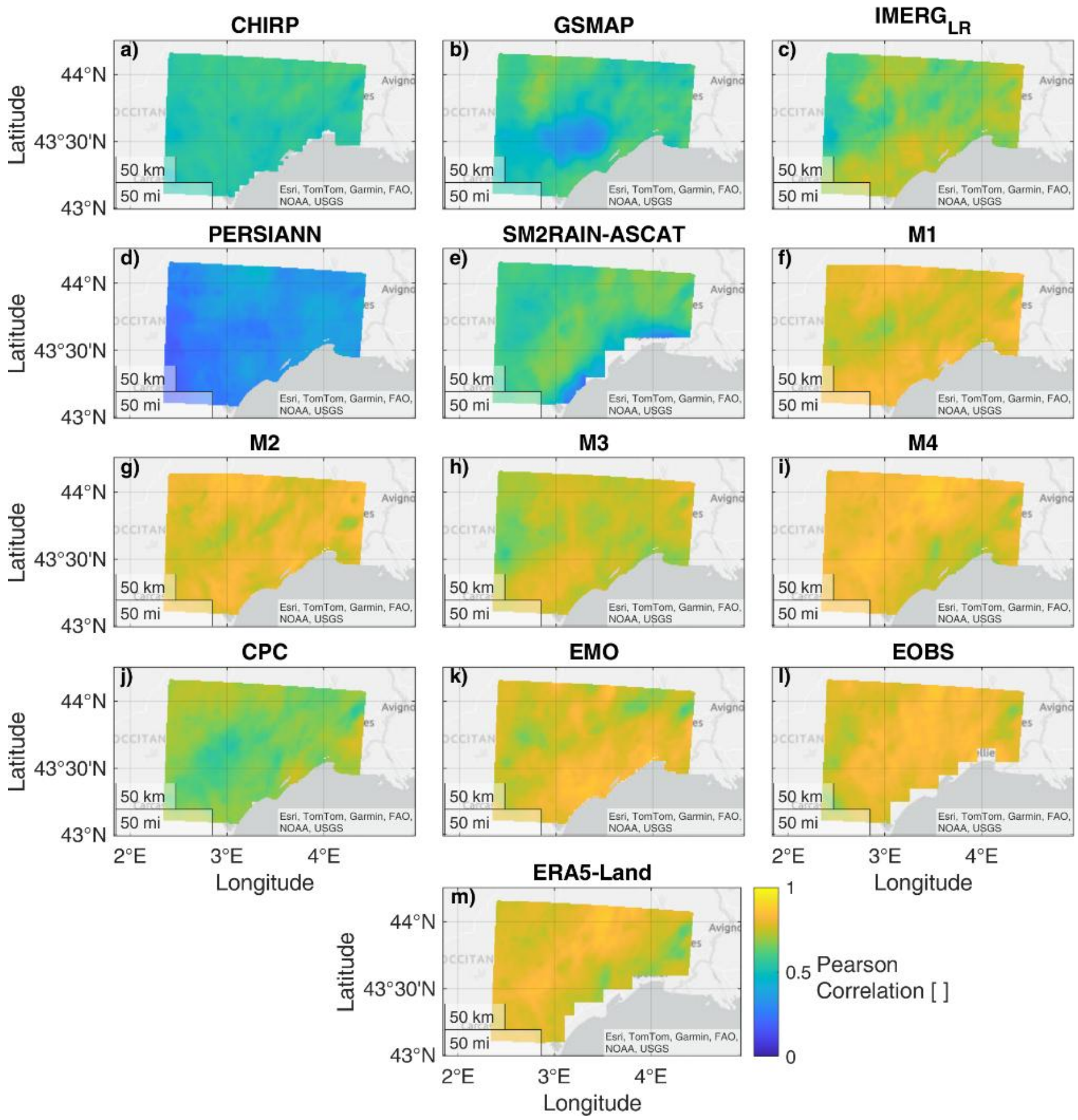

**Figure B2: Daily temporal Pearson's correlation of the selected precipitation datasets against COMEPHORE observations for the Hérault River basin area.**

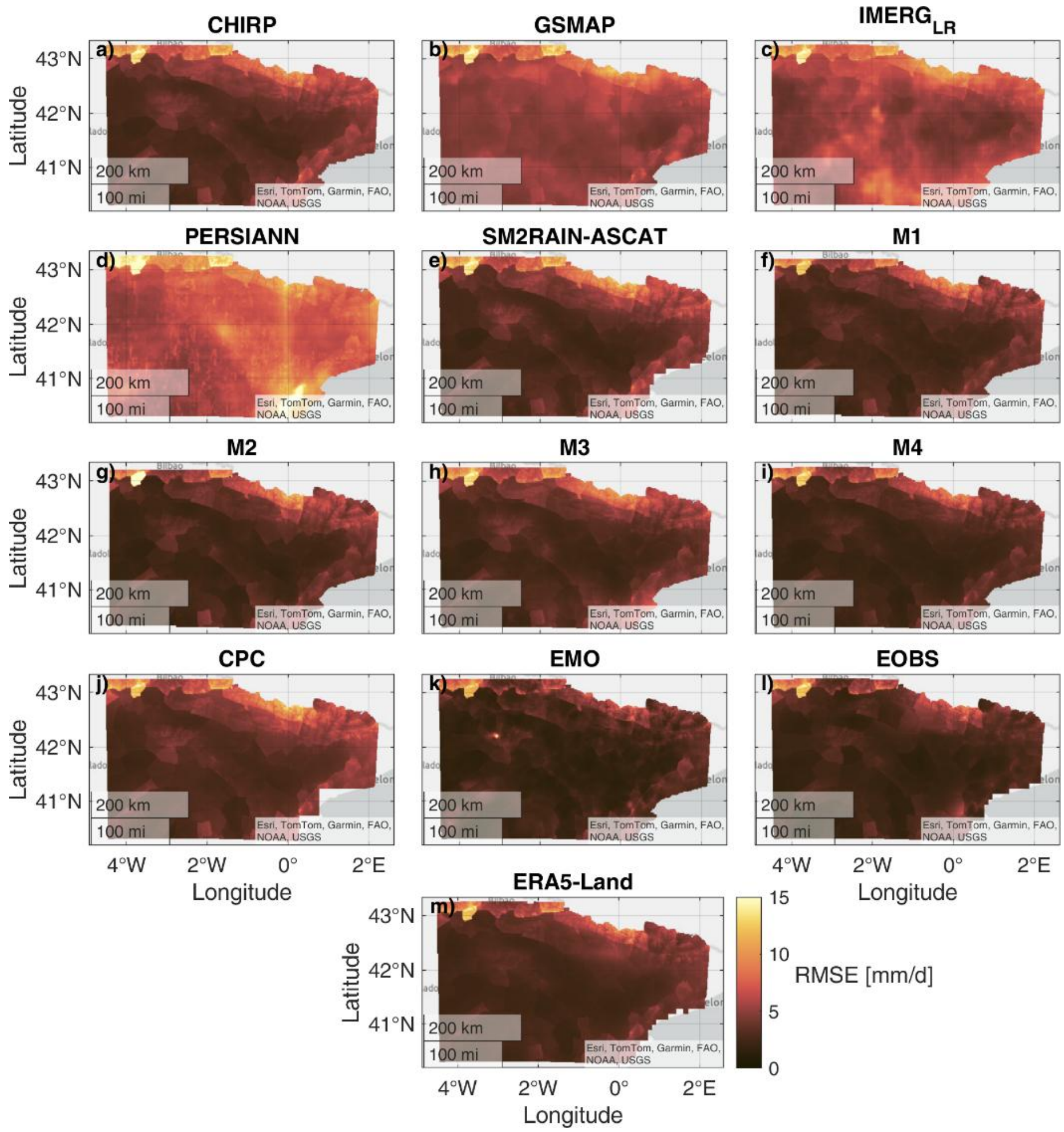

**Figure B3: Root Mean Square Error (RMSE) of the selected precipitation datasets against SAIH observations for the Ebro River basin area.**

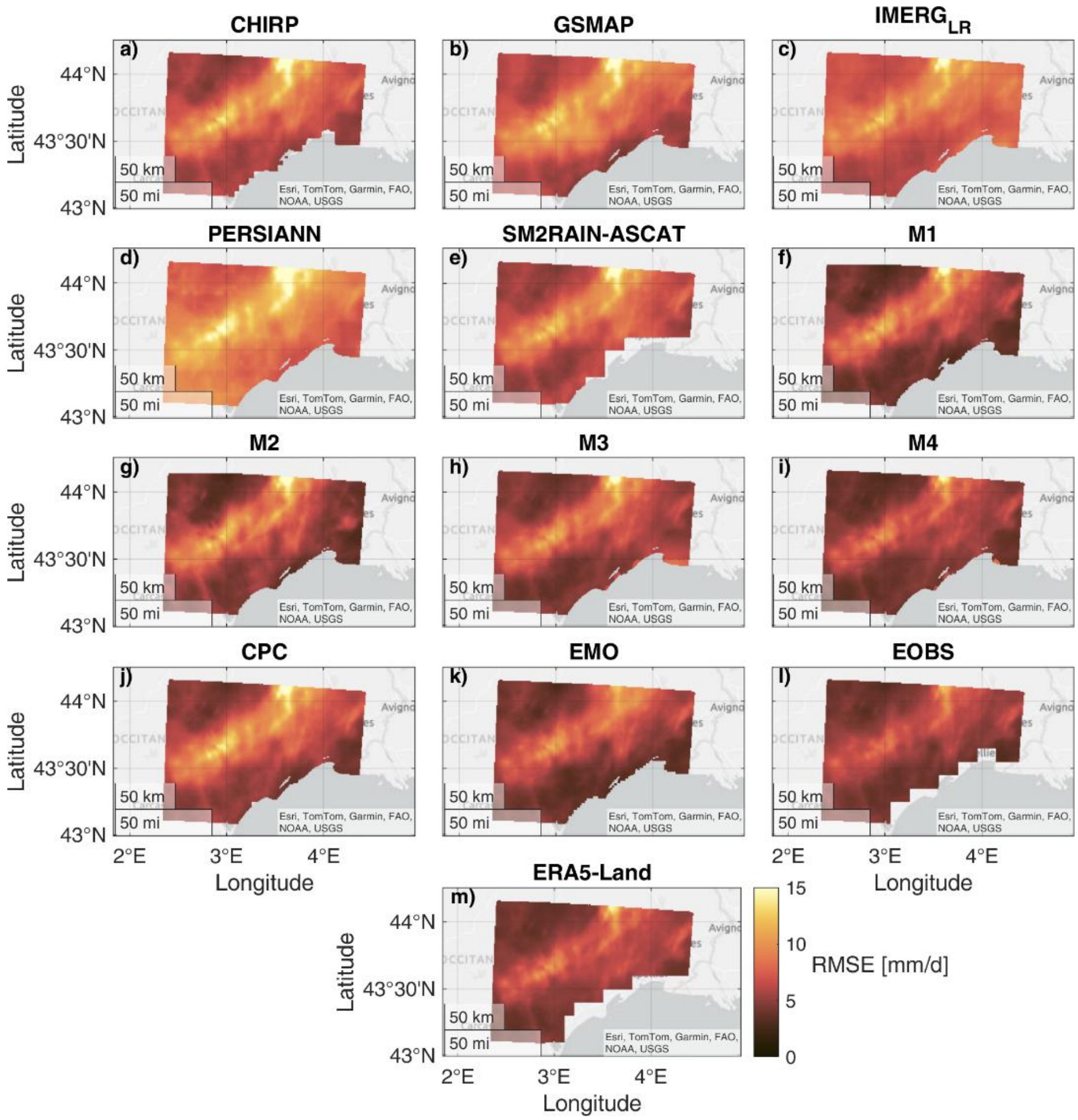

**Figure B4: Root Mean Square Error (RMSE) of the selected precipitation datasets against COMEPHORE observations for the Hérault River basin area.**

760

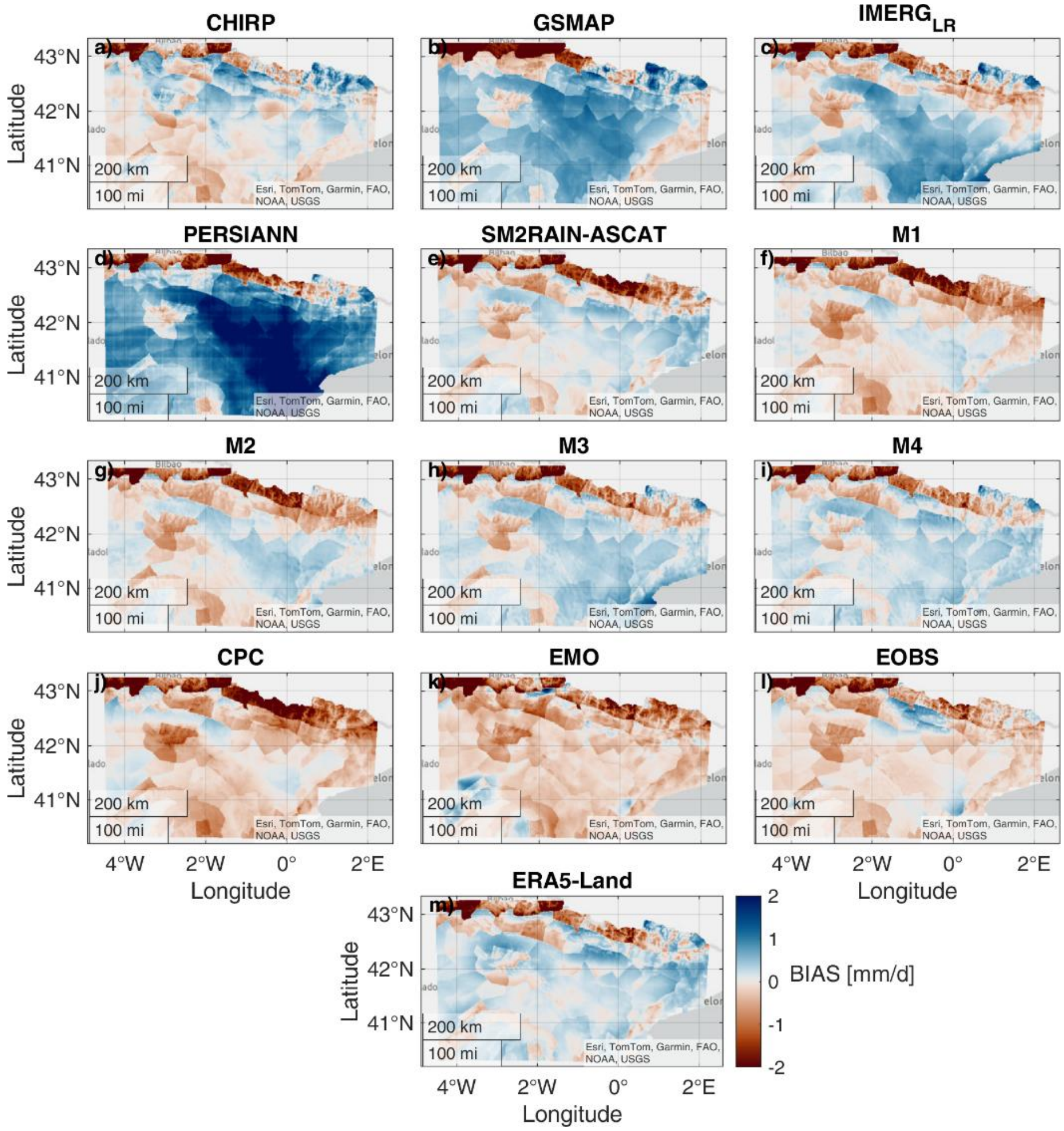

**Figure B5: BIAS error of the selected precipitation datasets against SAIH observations for the Ebro River basin area. Blue area means that the precipitation product overestimate precipitation, while brown area means underestimation.**

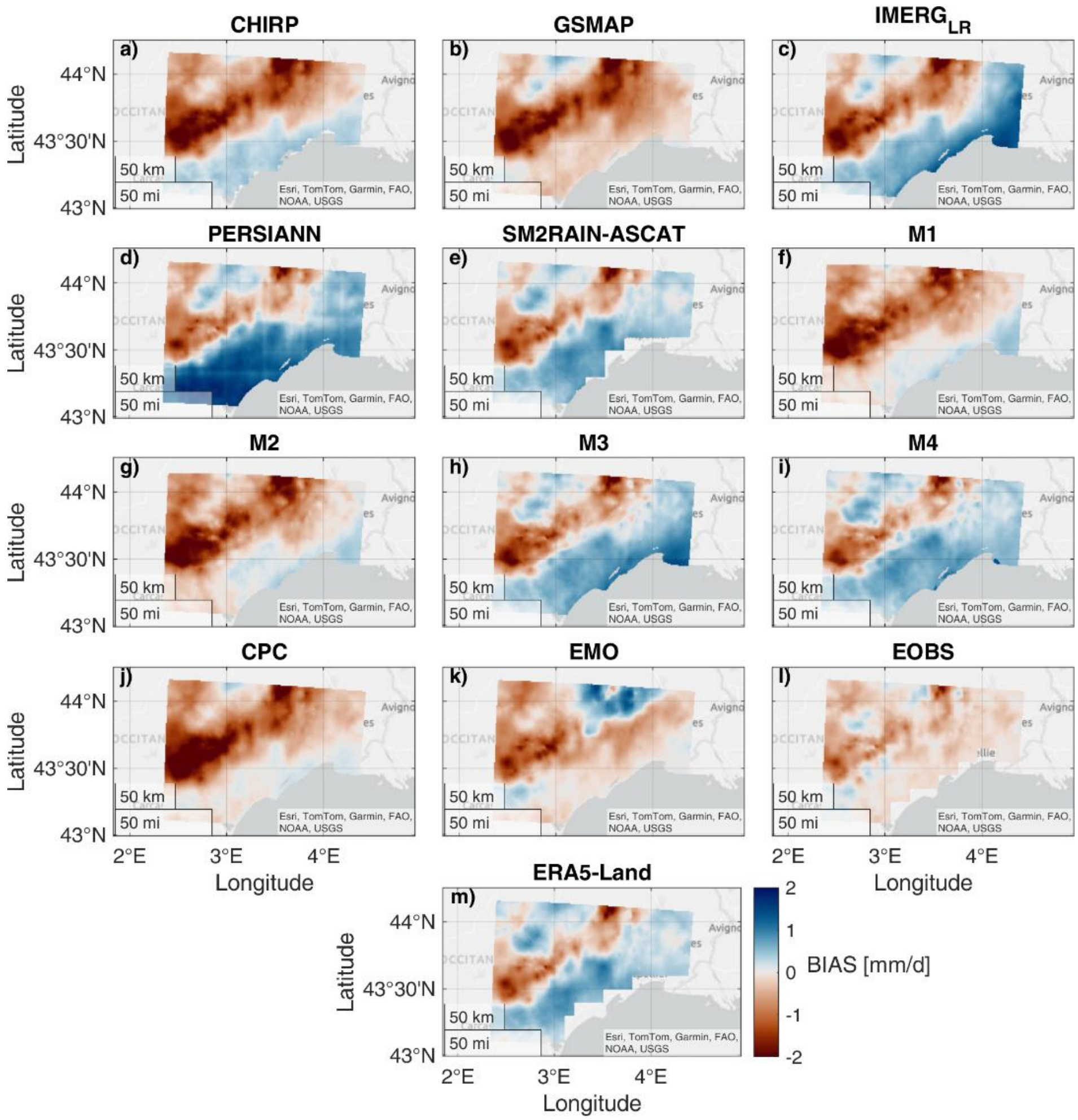

765

**Figure B6: BIAS error of the selected precipitation datasets against COMEPHORE observations for the Hérault River basin area. Blue area means that the precipitation product overestimate precipitation, while brown area means underestimation.**

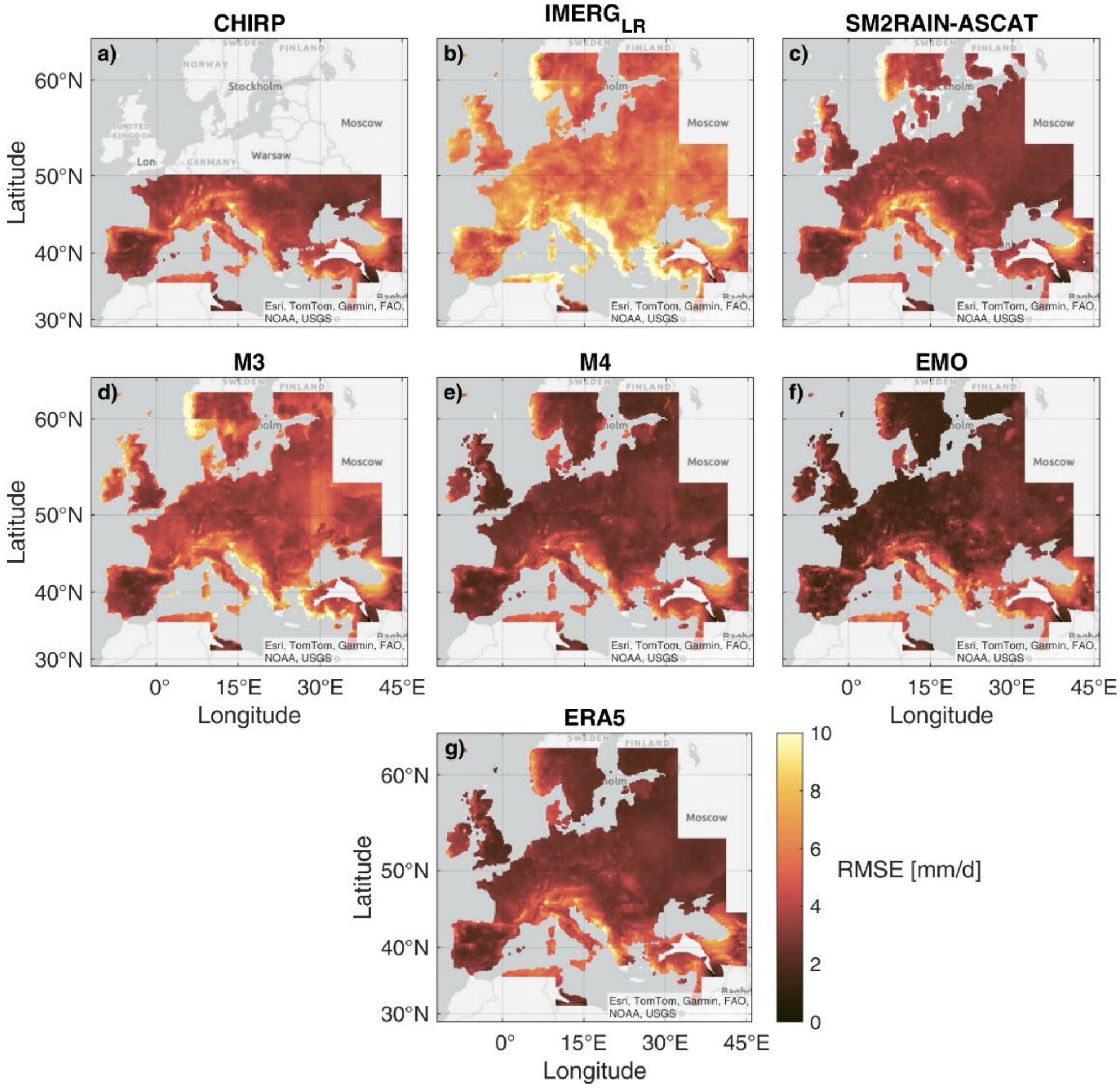

**Figure B7: Root Mean Square Error (RMSE) of the selected precipitation datasets against E-OBS observations for the full study area. All the datasets are aggregated at 10 km spatial resolution.**

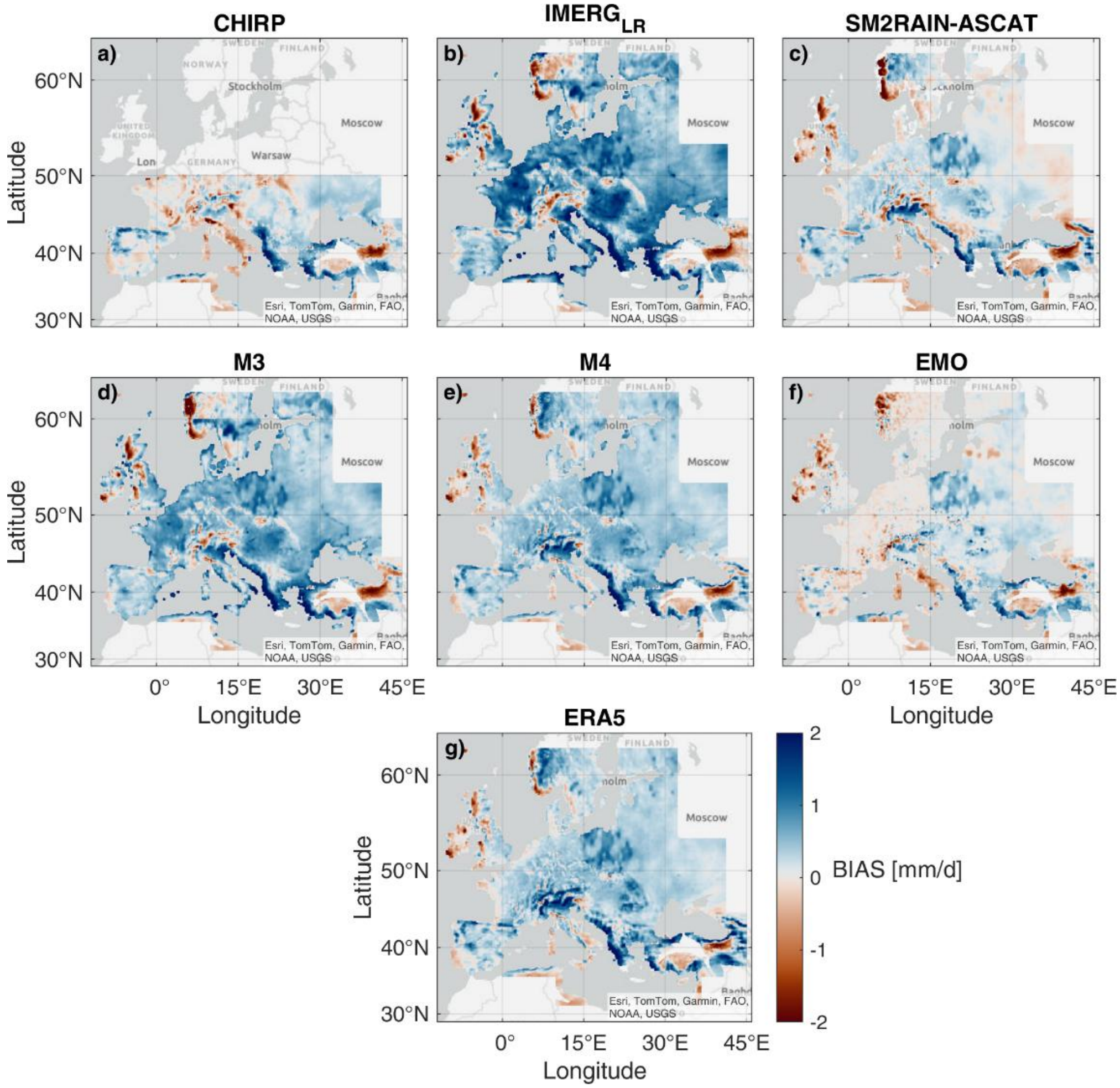

**Figure B8: BIAS error of the selected precipitation datasets against E-OBS observations for the full study area. Blue area means that the precipitation product overestimate precipitation, while brown area means underestimation. All the datasets are aggregated at 10 km spatial resolution.**

## Author contribution

PF: Conceptualization, Data curation, Formal analysis, Methodology, Software, Validation, Visualization, Writing – original draft preparation, Writing – review & editing; LB: Conceptualization, Funding acquisition, Supervision, Writing – review & editing; LC: Conceptualization, Resources, Writing – review & editing; HM: Investigation, Writing – review & editing; FA: Data curation, Investigation, Resources, Writing – review & editing; CM: Conceptualization, Funding acquisition, Methodology, Supervision, Writing – original draft preparation, Writing – review & editing

## Competing Interest

The authors declare that they have no conflict of interest.

## Acknowledgements

The work is supported by the European Space Agency (ESA) through the 4DMED-Hydrology project (grant no. ESA 4000136272/21/I-EF), the 4DHydro project (grant no. ESA 4000136272/21/I-EF) and Bridges project (bilateral agreement CNR/MHESR TUNISIA - CNR - MHESP Tunisia Agreement, B93C25000160001). The authors would like to thank Khaoula Khemiri and Anis Chkirbene for the support in validating the precipitation dataset over Tunisia.

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
