# Peer review of "Development of HYPER-P: HYdroclimatic PERformance-enhanced Precipitation at 1 km/daily over the Europe-Mediterranean region from 2007 to 2022"

_Earth System Science Data, 2025_

## Author Response (AR1)

We thank all the reviewers for their work in reviewing our paper. We greatly appreciated your comments, and we believe that they have helped us to improve our manuscript. Please find our replies below.

The reviewers' comments are shown with regular characters. Our replies are shown in bold. Text citations are shown in *italics*: bold characters highlight new additions, while removed text is . Please, note that all text citations refer to the revised manuscript with tracks.

Reviewer-1

I reviewed the manuscript "Development of HYPER-P: HYdroclimatic PERformance-enhanced Precipitation at 1 km/daily over the Europe-Mediterranean region from 2007 to 2022". A large number of (satellite) datasets is used aiming to identify the best precipitation dataset for (regions in) Europe and the Mediterranean region. Merged precipitation datasets outperform other datasets. The number of source and reference datasets is extensive and a detailed description of results is provided. It seems quite surprising that the merged datasets using rain gauge data, M1 and M2, are not the best performing datasets.

**We thank the reviewer for their valuable comments, which contribute to improve our manuscript quality and readability. Regarding the results of M1 and M2, this is probably due to the use of CPC as rain gauge product, which is characterized by coarse spatial resolution (0.5°) and limited number of gauge stations (around 17'000 for the full world: while E-OBS has ~14'000 just for Europe). We clarified this in the text (Lines 608-609 of the revised manuscript with tracks):**

*"**This is probably due to the native coarse resolution of the CPC dataset used in configuration M1 and M2, as well its relatively low number of included rain gauges**"*

**Indeed, if we had included the EOBS or EMO product, the performance might have improved. However, we would like to maintain the possibility to extend the data to other regions where these products are not available.**

**Additionally, the two datasets are precious for assessing the quality of the merged product, so their inclusion in it have made impossible to understand its performance. Please see below the reply to the remnant comments.**

Major comments

1) L. 93-95: Not much attention is given to radar-based datasets, although some of the

employed local reference datasets contain radar data. Indeed, the radar network across Europe is heterogeneous. I also agree that radars "are prone to errors", or better "sources of errors", but satellite products probably have more issues (as confirmed by the description of satellite limitations in the introduction). Not only in terms of accuracy, but also spatial and temporal resolution generally being lower compared to radar data (the latter being less relevant at the daily timescale). Ground-based radar seems a bit underrated in your study, whereas national meteorological and hydrological services typically employ ground-based radar and rain gauge data, sometimes using satellite data to remove non-meteorological echoes. Indeed, rain gauge data are typically needed to improve the radar precipitation estimates to a high enough level. Radars provide a more direct precipitation retrieval compared to most satellite sensors (radiometers, VIS, IR). On the other hand, satellites are often the only available source of information, apart from sparse rain gauge networks that sometimes only provide near real-time information at best, and hence definitely valuable also because long archives are available. Probably, combining radar, satellite and other datasets is the ultimate way forward, especially for areas without ground-based radar coverage or at far range from ground-based radars and rain gauges, where satellites can act as gap fillers.

Though the presented study is valuable to assess the performance of merged satellite-based datasets, which is also relevant for areas outside Europe (with similar climates) with sparse radar and rain gauge coverage, the importance of ground-based radars (merged with rain gauge data) for precipitation monitoring across Europe should be emphasized given its accuracy and generally higher spatiotemporal resolution compared to satellite data. Especially some outlook and recommendations concerning the combined use of rain gauge, ground-based radar, and satellite data (and possibly reanalysis data) would be appropriate. Note that several countries have publicly available climatological or real-time gauge-adjusted radar precipitation datasets (e.g., Germany, the Netherlands). And more high-value datasets across Europe are expected to become available in the future. At the European level, unadjusted OPERA radar products (not yet publicly available, but available for research purposes), or even OPERA-based gauge-adjusted radar products exist: a (near) real-time gauge-adjusted radar product (not yet publicly available; Park et al., 2019), a gauge-adjusted climatological radar product EURADCLIM (publicly available; Overeem et al., 2023). Such datasets have been compared to other (satellite-based) datasets (van der Plas et al., 2024; Lombardo and Bitting, 2024; Hammoudeh et al., 2025) and such references may even be used to put your results into perspective and add the "radar perspective". Now, readers may think that the best precipitation information over Europe comes from satellite-based precipitation products and that these products should always be used for near real-time or climatological purposes. I expect that gauge-adjusted radar precipitation products will generally give better estimates (but not everywhere and always, especially not in areas with lower rain gauge network densities), although they also have their own issues, and hence more research and development is needed to further improve these radar-based products (at the European scale also by combining with e.g. satellite and reanalysis data). It would also help in this respect to make clear what kind of datasets are useful for which applications. For instance, radar precipitation products are valuable in real-time

(hazardous weather warnings, input for nowcasting), whereas several satellite products, and hence HYPER-P, are not available in real-time (near real-time or climatological applications). This would help the reader to understand which datasets are available for which latency and hence application. Finally, I can also understand that limitations in availability of (gauge-adjusted) radar data hinder their actual use for new (satellite-based) products or applications.

**We completely agree with the reviewer. Indeed, Radar observation are far more accurate than satellite in both spatial and temporal resolution. For this reason, we selected them as benchmark. However, our objective is to create a stable product which can be potentially upscaled to other regions, hence we did not consider the merging with radar measurements. We underlined these concepts and the importance of radar measurements for Europe in the introduction, also highlighting the expected potential latency and applications for HYPER-P (Lines 93-101; 136-141):**

*"This technology allows for the collection of reliable precipitation information with high temporal resolution (often in the range of minutes), wide spatial coverage (a single weather radar can cover a circular area with a radius of 100-250 km) and better spatial accuracy compared to rain gauges, which only measure precipitation at their exact location). Radars do not measure rainfall directly; instead, they detect the reflectivity of precipitation particles. Radar measurements are hence often combined with rain gauges, to intercalibrate the measurements and obtain more reliable precipitation estimates.* However, also this network is heterogeneous in hardware, signal processing, frequency and scanning strategy, therefore their combination is difficult and prone to errors. **Moreover, most of the existing weather radars are mainly located in developed countries (Heistermann et al., 2013)."**

*"This product is generated by downscaling and merging multiple precipitation datasets from different sources: rain gauges, satellite observations (using both top-down and bottom-up approaches), and reanalysis data. The parent datasets are selected based on criteria such as low latency availability or potential, broad spatial coverage, and high accuracy. As a result, the merged product can be made available globally with relatively short latency—approximately one week. Radar measurements are not included in the merging process due to the lack of a global radar dataset and the limited number of weather radars, particularly in developing countries, but they can be used as a valuable reference."*

2) L. 270-274: The downscaling procedure needs more explanation. First, monthly CHELSA values are averaged to obtain standard year estimates. I assume these data are from the same year as the satellite data to which the downscaling will be applied, which is repeated for each year. Or do you mean that a climatological monthly average value is computed for each pixel? In addition, could you clarify this statement: "The monthly averages are then linearly

interpolated temporally for each day of the year". First, "standard year estimates" are obtained, I assume these are "the montly averages". But how do you interpolate the monthly average value temporally for each day of the year? Do you just divide by the number of days? And "standard year estimates" seems to average all monthly values, hence possible seasonal variation is not taken into account in the downscaling procedure. And why don't you use the daily CHELSA values, which would avoid this interpolation and seem to be available from the CHELSA portal? Finally, CHELSA data up to and including the year 2019 were downloaded, whereas the merged datasets run till 2022, suggesting that CHELSA is "only" used to derive average precipitation patterns, not necessarily based on data from the same period as the satellite data.

**We thank the reviewer for pointing out the lack of clarity in the description of the downscaling procedure. Since CHELSA dataset is not available in real time, and is only provided up to June 2019, we decided to use a single climatology for the full period: first, monthly aggregates were transformed in daily average by dividing each of them by the number of days in the respective month. Then, CHELSA estimate relative to the same calendar month were averaged to obtain 12 maps, each representative of a different month, thus creating a "standard year". The average monthly values of the standard year were then attributed to the central day of each month of the study period. A linear interpolation was applied to obtain a distribution estimation for each day of the study period, to avoid step-like patterns after the downscaling. We acknowledge that such strategy does not allow for a precise downscaling of the single storm patterns, due to the absence of concurrent high-resolution data. However, we expect that the high-resolution spatial patterns, even if averaged for each month, are useful to better spatialize coarse resolution information. Additionally, although CHELSA daily estimates are available on the website, they are limited to the year 1980 and were therefore not used. Since concurrent high-resolution data are not available for the full period, the influence of daily patterns on the final downscaled product is expected to be limited. The manuscript has been revised to clarify these aspects (Lines 289-297; 309-311).**

*"Since CHELSA dataset is not available in real-time but only up to June 2019, a single standard year climatology was used for the full period:* *first,* *m*onthly **aggregates**  **were converted into average daily precipitation by dividing each of them by the number of days in the corresponding month. Then, CHELSA estimates for the same month across 2000-2019** *we*re averaged to obtain **12 maps, each representative of a different month, thereby producing the** standard year estimates  (Fig. 1a). **The average monthly values of the standard year were then attributed to the central day of each corresponding month of the study period. Linear interpolation was then applied to obtain a daily estimate across the entire study period, thus avoiding step patterns after the downscaling***."*

*"Although this strategy does not allow a precise downscaling of the single storm pattern, due to the absence of a concurrent high-resolution pattern, it is useful to better spatialize coarse resolution information through the year"*

Minor comments

* L. 92-93: "Radar measurements are also increasingly available in Europe, with more than 700 operational weather radars managed by the EUMETNET within the OPERA". In the respective article it is stated that "The operational weather radar network in Europe covers more than 30 countries and contains more than 200 weather radars.". So "700" needs to be replaced by the still representative "200".

**We thank the reviewer for noticing the error. We have corrected the number.**

* L. 152-153: "and high-altitude regions, such as Scandinavia and parts of the Alps, with cold winters, short summers, and low precipitation": As can be seen in datasets, such as E-OBS, precipitation is generally high in the Alps and very high in parts of Scandinavia (at least Norway).

**We thank the reviewer for pointing it out. Indeed, Alps and Norway are characterized mainly by Dfc and ET Koppen climate, but due to the orographic lift effect, the precipitation is higher than usual. This has been opportunely clarified in the text (Lines 165-167):**

*" Subarctic **and Polar** Climates (Dfc, ET), present in Northern Europe and high-altitude regions, such as Scandinavia and parts of the Alps, with cold winters, short summers, and **generally** low precipitation**, even if the orographic lift in some cases affect the precipitation pattern (**Bonacina et al., 1945**)"***

* Table 1: The (catchment) size could be added or mentioned in the main text. In addition, the number of rain gauges or the gauge network density could be mentioned for the datasets employing rain gauges.

**We thank the reviewer for pointing it out. We added in the text the required information, where available, to the reference datasets description of section 2.2.1**

* I miss a map that shows the study areas (all regions).

**We have added a map of the selected cumulated precipitation product HYPER-P which clearly show the study area in the text (Figure 16)**

[Figure]

**Figure 16: Cumulated annual precipitation map of HYPER-P precipitation product, obtained by downscaling and merging precipitation product from reanalysis, satellite TD and satellite BU: specifically, ERA5-Land, IMERG Late Run and SM2RAIN-ASCAT.**

\* Are the employed satellite precipitation datasets completely independent from rain gauge data? For instance, perhaps IMERG-L is slightly dependent on rain gauge data (monthly bias adjustment) from a previous period?

**Thanks for asking: no, we have specifically selected products that do not incorporate rain gauge correction. Starting from version 05, IMERG applies the BIAS correction only in Final Run (https://gpm.nasa.gov/resources/faq/what-are-differences-between-imerg-early-late-and-final-runs-and-which-should-be-used). The other products were also checked. This will be specified by adding the sentence "*The selected version does not use any rain-gauge data*" to each of the products in section 2.2.2.**

\* L. 214: "combined with the globally gridded satellite from NOAA": add the exact product name.

**We have updated the description. Now it states (Lines 229-231):**

*"CHIRP  **combines** Thermal Infrared satellite precipitation estimates **from**  the G̶globally G̶gridded S̶satellite **(GriSat) and the Climate Prediction Center dataset (CPC TIR)** from NOAA to produce the precipitation dataset"*

* L. 244: "were linearly interpolated each day at midnight UTC": are all the daily values for all datasets in Table 1 from 0-24 UTC? For instance, for E-OBS it is known that gauge measurement intervals differ between networks. Could differences in actual daily measurement intervals affect your results?

**We thank the reviewer for highlighting this issue. We confirm that all the datasets, except for E-OBS, were originally available in UTC time. Indeed, we had overlooked the standard time convention used in E-OBS, given its role as a reference product and the relatively small temporal variation across Europe. However, following the reviewer suggestion, we have re-extracted the data and corrected the interpolation accordingly. As a result, EOBS performance has improved slightly, particularly in the coarse-resolution analysis, where regions with larger temporal discrepancies between standard time and UTC are included. Nevertheless, the overall results remain unchanged. We believe that any residual uncertainties related to the interpolation are minimal.**

* You could consider given normalized metrics to increase comparability of results between regions. For instance, the relative bias (bias divided by the mean rainfall of the reference) and the coefficient of variation of the residuals (standard deviation of the residuals divided by the mean rainfall of the reference).

**We thank the reviewer for their comment. Indeed, the relative indices are useful, but they risk overestimating small errors in arid areas and underestimating large errors in wet areas. Hence, we added a figure of cumulative precipitation for the full study area (Figure 16) to help the reader comparing the results in different region**

* L. 458: Please modify this sentence: "BIAS maps (Fig. 7", something went wrong in the description.

**We thank the reviewer for the comment. We will modified the sentence. Now it states (Lines 483-487):**

*"BIAS maps (Fig. 7)* ***show that all satellite-based precipitation products derived from the TD approach significantly underestimate precipitation in these regions, whereas SM2RAIN-ASCAT tends to overestimate it*** *"*

\* L. 542-543: "The absence of gauge stations in this area limits the reliability of both EMO and E-OBS products, which are forced to estimate precipitation data from sources other than rain-gauges and radar": which sources other than rain-gauges and radar are employed for E-OBS?

**We thank the reviewer for the comment. E-OBS uses topography information together with interpolation technique to better spatialize the precipitation estimates. We clarified this in the text (Line 571):**

*"The absence of gauge stations in this area limits the reliability of both EMO and E-OBS products, which are forced to estimate precipitation data from sources other than rain-gauges and radar **(e.g. topography information, Cornes et al., 2018).**"*

\* I find the suggestion to use and evaluate the merged precipitation products by computing discharges and comparing these to observed discharges is valuable from the use case perspective (and as pointed out by including references for some of the merged products).

**We thank the reviewer for the comment, which validates our analysis**

\* The choice for IMERG-L, being available in near real-time, suggests that the merged datasets are also relevant for near real-time applications. I suggest to briefly describe the purpose of the presented merged datasets, and especially HYPER-P: for near real-time and climatological applications or for climatological applications only given the latency of HYPER-P? Perhaps I missed it, but could you describe the expected update frequency and latency of HYPER-P?

**The expected latency was pointed out in line 335 (**"The objective product should be available with a relatively low latency (e.g., a week)"**). We further explained this point in the introduction to clarify it, also highlighting its potential applications (Lines 137-139; 144-145):**

*"**The parent datasets are selected based on criteria such as low latency availability or potential, broad spatial coverage, and high accuracy. As a result, the merged product can be made available globally with relatively short latency—approximately one week**"*

*"**Due to its potential low latency, HYPER-P can be useful for climatological applications like hydrological modeling, agricultural and drought monitoring or climatological studies.**"*

**Regarding the frequency update, we expect to continue working on this dataset to improve it and extend it for other areas. We highlighted this concept in the conclusions (Lines 690-691)**

*"Efforts will also aim to extend the dataset both spatially and temporally, especially in areas with sparse rain gauge coverage"*

References

* Hammoudeh S, Goergen K, Belleflamme A, Giles JA, Trömel S and Kollet S (2025) Evaluating precipitation products for water resources hydrologic modeling over Germany. Front. Earth Sci. 13:1548557. doi: 10.3389/feart.2025.1548557

* Lombardo, K., and M. Bitting, 2024: A Climatology of Convective Precipitation over Europe. Mon. Wea. Rev., 152, 1555–1585, https://doi.org/10.1175/MWR-D-23-0156.1.

* Overeem, A., van den Besselaar, E., van der Schrier, G., Meirink, J. F., van der Plas, E., and Leijnse, H.: EURADCLIM: the European climatological high-resolution gauge-adjusted radar precipitation dataset, Earth Syst. Sci. Data, 15, 1441–1464, https://doi.org/10.5194/essd-15-1441-2023, 2023.

* Park, S., Berenguer, M., and Sempere-Torres, D.: Long-term analysis of gauge-adjusted radar rainfall accumulations at European scale, J. Hydrol., 573, 768–777, https://doi.org/10.1016/j.jhydrol.2019.03.093, 2019

* van der Plas, E., A. Overeem, J. F. Meirink, H. Leijnse, and L. Bogerd, 2024: Evaluation of IMERG and MSG-CPP Precipitation Estimates over Europe Using EURADCLIM: A Gauge-Adjusted European Composite Radar Dataset. J. Hydrometeor., 25, 1177–1190, https://doi.org/10.1175/JHM-D-23-0184.1.

Reviewer-2

I have reviewed the manuscript 'Development of HYPER-P: HYdroclimatic PERformance-enhanced Precipitation at 1 km/daily over the Europe-Mediterranean region from 2007 to 2022'. I found the datasets well described and useful, including analysis and discussion on performance, uncertainties and applications. I have no further comments.

**We thank the reviewer for their comment, which validates our analysis**

---

## Author Response (AR2)

**We thank the reviewer for their work in reviewing again our paper. We have revised our paper according to their suggestion. Please find our replies below.**

**The reviewers' comments are shown with regular characters. Our replies are shown in bold. Text citations are shown in *italics*: bold characters highlight new additions, while removed text is . Please, note that all text citations refer to the revised manuscript with tracks.**

Reviewer-1

In general, the authors responded very well to my review comments. My only remaining concern is related to the use of E-OBS for comparison of daily precipitation. In addition, I have a few minor comments. Line numbers refer to the manuscript with track changes.

Main point
Regarding the interpolated rain gauge dataset E-OBS, you state in your rebuttal: "We confirm that all the datasets, except for E-OBS, were originally available in UTC time. Indeed, we had overlooked the standard time convention used in E-OBS, given its role as a reference product and the relatively small temporal variation across Europe. However, following the reviewer suggestion, we have re-extracted the data and corrected the interpolation accordingly. As a result, EOBS performance has improved slightly, particularly in the coarse-resolution analysis, where regions with larger temporal discrepancies between standard time and UTC are included."

My point is that E-OBS does effectively not provide an accumulation for that day from 0-24 UTC, because of the different measurement intervals of rain gauge networks, especially from different countries, but often also within countries. The time stamp of data from different networks is mostly in UTC, but the measurement interval often differs. This is at least described in Section 2.2 in Overeem et al. (2023), with, for instance, end times of observation of 06:00 UTC and 18:00 UTC. This is important given the many figures where daily precipitation is compared. I expect that this will not be important for bias computations over longer periods. Timing differences will be negligible then. But I do expect that this will influence the results for other metrics.

In addition, what do you mean by "we have re-extracted the data and corrected the interpolation accordingly"? How to achieve this for E-OBS, since this is already a gridded dataset? And does this imply that you took into account the above mentioned differences in measurement interval? Or did you take this into account by selecting the appropriate satellite data? This is difficult to disentangle, because this would require different selections per country, and measurement intervals even differ within countries.

**We thank the reviewer for the clarification. Our initial intervention was related to the fact that E-OBS aggregations are available at local time rather than UTC. Therefore, we interpolated E-OBS data to align the UTC time zone, with slight performance improvement in Eastern Europe areas.**

**Indeed, the issue arising from E-OBS combining data with different temporal aggregation introduces uncertainty in the evaluation of precipitation products—an inherent limitation that cannot be easily resolved. However, since E-OBS is just one of several datasets used in the assessment, and given its widespread use in the scientific community, we have chosen to retain it as a reference dataset while clearly informing readers about the associated uncertainties (Lines 209–214):**

*"Note that, in some areas, E-OBS observations are derived by aggregating precipitation stations with time intervals that differ from the standard 00–24 period (Overeem et al., 2023), This can potentially cause uncertainty in the assessment of precipitation products using E-OBS. However, considering that E-OBS is not the only dataset used as reference and the importance of assessing HYPER-P against widely used precipitation products, the uncertainty is deemed acceptable."*

I invite the authors to provide an explanation to address this concern. Perhaps I'm overlooking something.

I have a few remaining (very) minor comments:
1) L. 97-98: "Radar measurements are hence often combined with rain gauges, to intercalibrate the measurements and obtain more reliable precipitation estimates.". I suggest to use "adjust" instead of "intercalibrate". In radar meteorology, "calibration" is typically used for hardware calibration.

**We thank the reviewer for the suggestion. We have changed the term accordingly.**

2) The additions in the text on the suitability of radars for precipitation estimation and the reason to not include their data are much better described by (L. 139-141): "Radar measurements are not included in the merging process due to the lack of a global radar dataset and the limited number of weather radars, particularly in developing countries, but they can be used as a valuable reference.". However, I recommend to be more explicit by adding/rewriting that:
- the main purpose of HYPER-P is to provide a "consistently processed" global dataset, especially geared towards application in (nearly) ungauged regions lacking radar coverage. Hence, the main goal is to improve precipitation data in those regions (although the dataset is also expected to add value for regions in Europe further/far away from radar sites and/or rain gauges).
- that some gauge datasets (e.g. EMO, E-OBS or individual timeseries), as well as radar data

have not been incorporated to keep an independent reference for verification above Europe, which is used as a testbed.

**We thank the reviewer for the suggestion. We had already pointed out the potential of HYPER-P to be global in its introduction (line 137-139):** "The parent datasets are selected based on criteria such as low latency availability or potential, broad spatial coverage, and high accuracy. As a result, the merged product can be made available globally with relatively short latency—approximately one week"
**We added two sentences to the text to highlight the above-mentioned topics:**

**Lines 141-143:** "*Local (intended as not-global) datasets from radar and gauge were not included in the merging, but they were used as independent references for assessing the performance of the merged product.*"

**Line 147-148:** "*Specifically, HYPER-P is expected to be particularly valuable for completely or nearly ungauged areas, which lack stable and high-resolution information from ground networks (gauges and/or radars).*"

3) Note that topography information is used in E-OBS, but not for the variable precipitation (I've checked this with an E-OBS expert).

**We thank the reviewer for the suggestion. We have corrected the related sentence in the manuscript. Now it states (Lines 572-574):**

"*The absence of gauge stations in this area limits the reliability of both EMO and E-OBS products, which are* **based on**  **spatial-interpolation techniques**  *Cornes et al., 2018).*"

Bibliography

Overeem, A., van den Besselaar, E., van der Schrier, G., Meirink, J. F., van der Plas, E., and Leijnse, H.: EURADCLIM: the European climatological high-resolution gauge-adjusted radar precipitation dataset, Earth Syst. Sci. Data, 15, 1441–1464, https://doi.org/10.5194/essd-15-1441-2023, 2023.

---

## Author Response (AR3)

Please ensure that the colour schemes used in your maps and charts allow readers with colour vision deficiencies to correctly interpret your findings. Please check your figures using the Coblis – Color Blindness Simulator (https://www.color-blindness.com/coblis-color-blindness-simulator/) and revise the colour schemes accordingly with the next file upload request. --> Fig. A2, A4

**Thanks for the suggestion. I had already changed the figure in the last review round. According to Coblis – Color Blindness Simulator, the current version of A2 and A4 is adapt for readers with colour vision deficiencies**